# A New Method for Calculating Highway Blocking due to High Impact Weather Conditions

Duanyang Liu[1], Tian Jing[1, 2, 6], Mingyue Yan[3], Ismail Gultepe[4], Yunxuan Bao[1,2,5], Hongbin Wang[1], Fan Zu[1]

1. Key Laboratory of Transportation Meteorology of China Meteorological Administration, Nanjing Joint Institute for Atmospheric Sciences, Nanjing, Jiangsu 210041, China.
2. Collaborative Innovation Center on Forecast and Evaluation of Meteorological Disasters,Nanjing University of Information Science & Technology,Nanjing 210041, China.
3. Highway Monitoring & Emergency Response Center, Ministry of Transport of P.R.C
4. Ontario Tech University, Oshawa, Toronto, ONT, Canada
5. Engineering Research Center for Internet of Things Equipment Super-Fusion Application and Security in Jiangsu Province, Wuxi University, Wuxi 214105, China.
6. Advanced Institute of Natural Sciences, Beijing Normal University, Zhuhai 519087, China;

*Correspondence to*: Duanyang Liu (liuduanyang@cma.gov.cn), Tian Jing  (jingtian819216471@163.com), Mingyue Yan (2514926295@qq.com)

**Abstract**: Fog, rain, snow, and icing are the high-impact weather events often lead to the highway blockings, which in turn causes serious economic and human losses. At present, there is no clear calculation method for the severity of highway blocking which is related to highway load degree and economic losses. Therefore, there is an urgent need to propose a method for assessing the economic losses caused by high-impact weather events that lead to highway blockages, in order to facilitate the management and control of highways and the evaluation of economic losses. The goal of this work is to develop a method to be used to assess the high impact weather (HIW) effects on the highway blocking. Based on the K-means cluster analysis and the CRITIC (Criteria Importance through Intercriteria Correlation) weight assignment method, we analysed the highway blocking events occurred in Chinese provinces in 2020. Through cluster analysis, a new method of severity levels of highway blocking is developed to distinguish the severity into five levels. The severity levels of highway blocking due to high-impact weather are evaluated for all weather types. As a part of calculating the degree of highway blocking, the highway load in each province is evaluated. The economic losses caused by dense fog are specifically assessed for the entire country.

## 1 Introduction

The World Meteorological Organization (WMO) defines high-impact weather as severe weather events that have significant adverse impacts on society, infrastructure, and the environment. These events can cause widespread damage, disruption, and loss of life (Marsigli, 2021). High-impact weather conditions negatively affect the transportation safety, mobility, and reliability (Das and Ahmed, 2022; Hammit et al. 2018; Dehman and Drakopoulos, 2017; Yu et al. 2015), even causing delays in the delivery of goods and materials (Jaroszweski et al., 2015). Many agencies have set enhancing traffic safety on

highways as the main goal to improve transportation efficiency and safety (Ali et al. 2021; Das et al. 2021; Ratanavaraha and Suangka, 2014).

Weather-related road accidents have a significant seasonal pattern as the high-impact weather is common in different seasons (Edwards, 1999; Keay and Simmonds, 2005; Bergel Hayat et al., 2013). Fog, heavy rain and snow usually have the most adverse impact on transportation (Yang et al. 2021). Due to slippery road conditions or hydroplaning on rainy days (Kim et al. 2021), road accidents are two to three times more likely than in dry weather (Brodsky and Hakkert,1988) and the overall accident risk was found to be 70% higher than the value of normal conditions (Andrey and Yagar, 1993). Fog usually occurs

in winter and has a great impact on transportation (Li et al., 2019; Liu et al., 2016; Shen et al., 2022). Low visibilities because of fog can affect the drivers' behaviour and the road safety (Hassan and Abdel-Aty, 2011), and vehicle rear-end collisions on the highway commonly occur in foggy weather (Huang et al. 2020). Therefore, driving in foggy weather is a potentially dangerous activity for road block (Yan et al. 2014).

The high-impact weather conditions can also lead to traffic delays, economic loss, or increased pollution effects (fog case).

The road surface with snow or icing can lead to slower vehicle speeds and a decrease in fuel combustion efficiency (Hallegatte 2008; Min et al., 2016). The work of Min et al. (2016) showed that when 10% improvement occurs in road surface conditions, 0.6–2% reduction in air emissions amount can occur. If the weather forecasting products are used, the road transportation sectors can generate great economic benefit. Frei et al. (2014) found that the use of meteorology in the road transportation sector in Switzerland generates an economic benefit to the national economy 75.1-91.2 million U.S.

dollars (cost/benefit ratio of around as 1:10).

Highway Blockings are usually caused by human or natural causes of highway traffic interruption, and they are generally divided into the two types (Niu et al. 2015; Song et al. 2021): 1) due to highway maintenance construction, reconstruction and expansion construction, major social activities and other planned events and 2: due to natural disasters (including geological disasters, extreme weather, etc.), accidents, disasters, public health events, social security events, and other

reasons caused by unexpected events.

Many researchers have studied the influence of high-impact weather on highway traffic blocking (Song et al., 2021; Niu et al., 2015). Song et al., (2021) found that the correlation coefficients between the length of highway blocking and the meteorological impact indices of fog, typhoon and snowfall are significantly positive. However, there are few studies on the damage caused by highway blocking due to high impact weather (HIW) (Andrey et al. 2003; Chapman 2015; Chen et al.,

2016; Manish et al., 2005; Pregnolato et al., 2017). In this respect, the current work will further help to improve HIW events prediction and the effects on road blocking. By 2021, the total length of highways in China has reached 169,100 kilometers, ranking first in the world (Liu et al., 2022). Therefore, there is a critical need to improve the ability to estimate highway traffic, caused by the highway blocking during high-impact weathers.

In this study, we describe and evaluate the main high-impact weather (HIW) components (fog, rain, snow, and icing), their

temporal and spatial distribution characteristics, highway loads, and economical losses caused by highway blocking. Based on this analysis, a new method for assessing the impact of HIWs on road blocking is proposed. The findings of this study

provide transport institutions with practical guidance for systematically investigating the effects of weather on highway blocking damage.

This manuscript is organized as follows: The data and methods are described in section 2. The highway blocking features, temporal and spatial distribution, and damage caused by the highway blocking are given in section 3. Discussions are given in Section 4 and the conclusions are provided in section 5.

## 2. Data and methods

### 2.1 Chinese highway network and study area

In this study, the highway distribution in 2020 issued by the Ministry of Transport of the People's Republic of China is adopted, with a total mileage of 155,000 km (Fig. 1a). According to the differences in geographical environment and climatic characteristics of China, 32 provincial administrative regions in China (Taiwan, Hong Kong and Macao are not included in this study due to the lack of data) are divided into 8 main regions (Fig. 1b), i.e., Northeast China (Heilongjiang, Jilin and Liaoning Provinces and north-eastern Inner Mongolia Autonomous Region), North China (Beijing and Tianjin Municipalities, Hebei, Shanxi and Shandong Provinces and central Inner Mongolia), East China (Jiangsu, Anhui and Zhejiang Provinces, and Shanghai Municipality), Central China (Henan, Hubei, Hunan and Jiangxi Provinces), South China (Guangxi, Guangdong, Fujian and Hainan Provinces), Northwest China (Xinjiang Autonomous Region, Gansu Province, Ningxia Hui Autonomous Region, Shaanxi Province and western Inner Mongolia), Qinghai-Tibet region (Tibet Autonomous Region and Qinghai Province), and Southwest China (Sichuan, Yunnan and Guizhou Provinces).

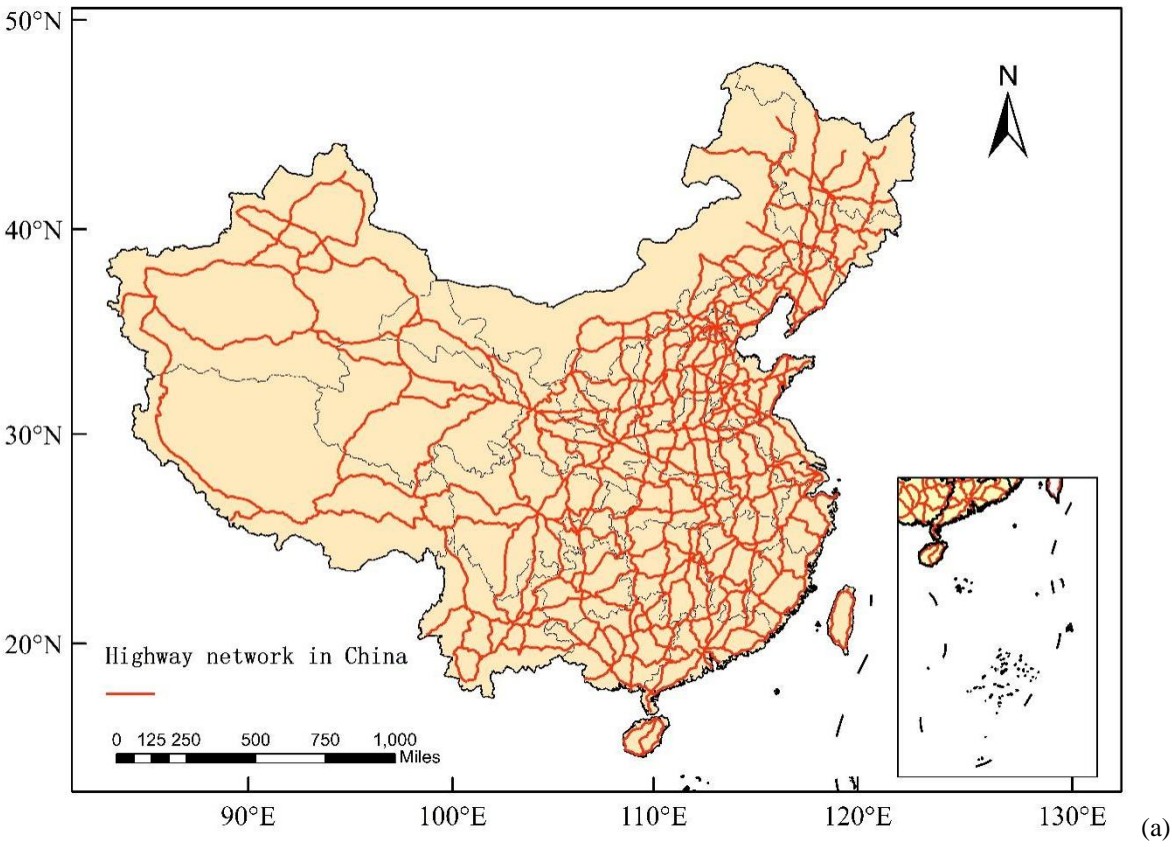

(a)

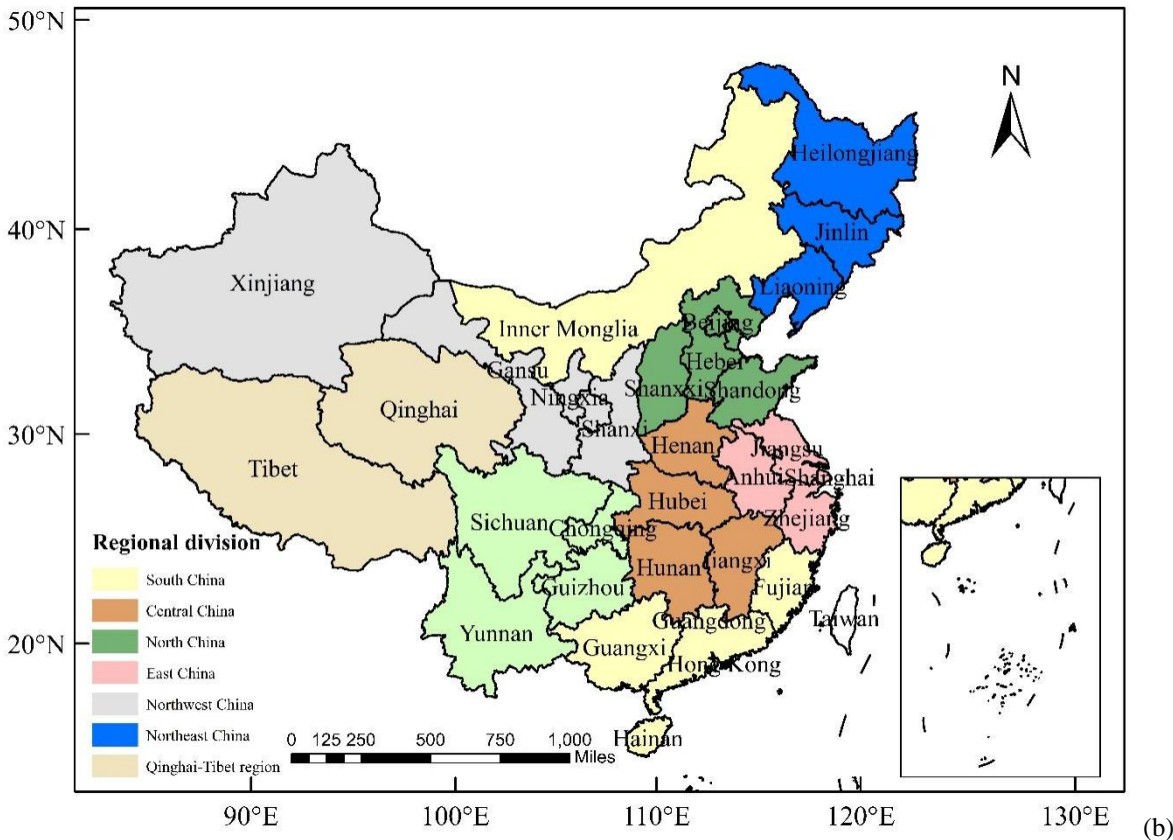

**Figure 1. The distribution of highway network (a) and the regional division (b) in China.**

## 2.2 Data sources

### 2.2.1 Highway-blocking events data

A highway-blocking events is a state in which a highway is impassable or forced to close for some reason. Depending on the nature and duration of the blockage, it can be divided into two categories: planned and unexpected. Planned blockages include those caused by planned events such as highway maintenance and construction, reconstruction and expansion, and major social activities. Unexpected blockages include sudden highway blockages caused by natural disasters (such as geological disasters, severe weather, etc.), accidents and disasters, public health incidents, social security incidents and other reasons. In this study, highway blockage under the influence of natural disasters and weather is selected as the main body of the study. The highway-blocking events data obtained from the Ministry of Transport of the People's Republic of China (Fig. 1a) follow the criteria of the Highway Traffic Blocking Information Submitting System of the Ministry of Transport of the People's Republic of China (2018, No. 451; https://www.hunan.gov.cn/xxgk/wjk/zcfgk/202007/t20200730_e1c6436a-6aff-43d0-9c74-c0822311b8db.html). Highway blockage data comes from Highway Monitoring & Emergency Response

Center, and the data recording process and specifications are in accordance with the "Information Reporting System for Highway Traffic Blockage of the Ministry of Transportation and Communications of the People's Republic of China" issued by the Ministry of Transportation and Communications of the People's Republic of China.The dataset contains 16 indicators: province name, submitting department, route name, route number, starting and ending pile number, reasons of highway blocking, blocking mileage, status, blocking type, information event classification, site description, disposal measure, time of finding blockage, submitting time, expected recovery time, and actual recovery time. (There are two nouns that need to be explained in detail: firstly, Highway pile numbers are usually combined with the milestone system and are expressed in K kilometers ± meters. That is, along the direction of the road, the pile number at the starting point is k0+000, and one pile number is marked every certain distance (such as 100 meters), and the corresponding place is marked. Second, the blocking mileage is the distance of the highways blocking, for example, due to flash floods caused by precipitation in mountainous areas, there are 100 kilometers of highway cannot be used normally, we assume that the blocked mileage caused by heavy rain is 100 kilometers). Since highway blocking information is submitted by manual statistics, there is a possibility of manual statistical errors. Therefore, all data were pre-corrected with a time series correction and then verified based on the cause of the blockage and the meteorological data of the station at the time, among other things. Quality control resulted in the retention of 95% of valid data.

### 2.2.2 Meteorological observations

The meteorological observation data used in this research is obtained from the National Climate Center of the China Meteorological Administration, and its variables include temperature, pressure, wind direction and speed, rainfall, snow depth and visibility. We designate March–May as spring, June–August as summer, September–November as autumn and December–February as winter. The weather cause of the blockage is recorded in the highway blocking data. We can find the relevant meteorological elements of the nearest weather station in this period according to the road section and time period of the blockage. According to the time and place of the expressway blockage, the meteorological observation data of this area during this period are checked with the weather events recorded by the observers to ensure the consistency of the data. In the case of multiple weather phenomena, we refer to the one recorded by highway blocking recorded data.

### 2.2.3 Transportation-related economic data

The transportation-related economic data are derived from the National Bureau of Statistics for transportation and post item and national economic accounting item (2020 data). The transportation and postal item contain the transport mileage, Highway Technology Status, passenger capacity, freight capacity, civilian automobile ownership and express business. The national economic accounting includes the added value of gross domestic product (GDP) generated by transportation. The above data are the traffic per 10 kilometers of the highway, per month(The industry classification of transportation is based

on the industry classification of national economic activities, https://data.stats.gov.cn/easyquery.htm?cn=C01). The above data will be calculated and applied in 2.3.3 Calculation method of highway load.

## 2.3 Analytical method

### 2.3.1 K-means clustering analysis

K-means clustering is an unsupervised machine learning method without prior knowledge (that is, no classification criteria is given before classification). The goal of this algorithm is to find groups in the data, with the number of groups represented by the variable K. One chooses the desired number of clusters, and the K-means procedure iteratively moves the centers to minimize the total within-cluster variance. Specifically, the criterion is minimized by assigning the observations to the K clusters in such a way that within each cluster the average dissimilarity of the observations from the cluster mean, as defined by the points in that cluster, is minimized (Hastie et al., 2009).

Consider a set of n-dimensional vector $\{x_i\} = \{x_{i1}, x_{i2}, \ldots, x_{in}\}$, and the dissimilarity measure follows the squared Euclidean distance.

$$d(x_i, x_{i'}) = \sum_{j=1}^{n}(x_{ij} - x_{i'j})^2 = \|x_i - x_{i'}\|^2 \tag{1}$$

The objective is to find

$$\text{argmin}\frac{1}{2}\sum_{k=1}^{K}\sum_{C(i)=k}\sum_{C(i')=k}\|x_i - x_{i'}\|^2 = argmin\sum_{k=1}^{K}N_k\sum_{C(i)=k}\|x_i - \bar{x}_k\|^2$$

(2)

where $N_k$ is the number of points in cluster $k$, $\bar{x}_k$ is the center of cluster $k$, $C(i) = k$ indicates $x_i$ belongs to cluster $k$. Given an initial set of centers, the K-means algorithm alternates these steps:

(1) Assigning points to clusters: for each point in the dataset, calculate its distance (commonly used Euclidean distance) from the centers of all clusters and assign the point to the cluster corresponding to the cluster center with the smallest distance. This step realizes the initial segmentation of the data points.

(2) Update cluster centers: for each cluster, recalculate the mean (i.e., center of mass) of all data points in the cluster and use this mean vector as the new cluster center. This step is the core of the algorithm, which makes the center of each cluster gradually approach the center of the true distribution of points within the cluster, thus optimizing the clustering results.

(3) Check the convergence condition: determine whether the cluster center has changed or whether the change is less than a certain threshold, or whether the preset maximum number of iterations has been reached. If the convergence conditions are met, the algorithm ends; otherwise, return to step 1 to continue iteration.

For each center we identify the subset of training points (its cluster) that is closer to it than any other center. The means of each feature for the data points in each cluster are computed, and this mean vector becomes the new center for that cluster. These two steps are iterated until convergence. Typically, the initial centers are randomly chosen observations from the

training data. Details of the K-means procedure, as well as generalizations allowing for different variable types and more general distance measures, are given in Hastie et al. (2009).

### 2.3.2 CRITIC weight assignment method

The CRITIC (Criteria Importance through Intercriteria Correlation) method is an objective weight assignment method (Diakoulaki et al. 1995; Wei et al. 2020), which is commonly used for the analysis of data with strong correlations of indicators and also considers the variability among indicators. By objectively calculating the indicators of data, each item is assigned a different weight, and the calculation steps are as follows.

Contrast intensity, expressed as a standard deviation, indicates the dispersion degree of an indicator. The larger the standard deviation is, the greater the dispersion degree is, the larger the differences between samples are, and the larger the assigned corresponding weights are. The standard deviation $S$ can be expressed in Eq. (3).

$$S_j = \sqrt{\frac{\sum_{i=1}^{p}\left(x_{ij}-\frac{1}{n}\sum_{i=1}^{n}x_{ij}\right)^2}{n-1}} \tag{3},$$

where $x_{ij}$ denotes the data processed by standard deviation, $x_{ij}$ represents the value of the $j$th evaluation index of the i sample, $S_j$ the standard deviation of the $j$th indicator, $n$ the total number of samples, and $p$ the total number of indicators.

Correlation is expressed as the correlation coefficient between indicators. The stronger the correlation between indicators is, the higher the repetition rate of information expression. Therefore, the corresponding weights of the indicators can be reduced to a certain extent. The correlation coefficient $R$ can be expressed in Eq. (4).

$$R_j = \sum_{i=1}^{p}(1-r_{ij}) \tag{4},$$

where, $R_j$ indicates the correlation coefficients of the $j$th indicator with the other indicators, and $r_{ij}$ denotes the correlation coefficient of the $i$th indicator with the $j$th indicator.

The weight of indicator ($W$) can be written as Eq. (5).

$$W = \frac{S_j \times R_j}{\sum_{j=1}^{p} S_j \times R_j} \tag{5}.$$

In the CRITIC weighting method, although the standard deviation, as a measure of the intensity of comparison, reflects the magnitude of the difference in the values of the same indicator between different evaluation objects, that is, the volatility, the magnitude of the standard deviation is not the only factor that directly determines the magnitude of the weights. The determination of weights also needs to consider the conflicting nature of different indicators, that is, the correlation between them. When an indicator has a large standard deviation and high correlation with other indicators, its weight may be weakened by a high correlation (i.e., low conflict). Conversely, when an indicator has low correlation with other indicators

(i.e., high conflict), although not the largest standard deviation, its weight may be relatively high because of the unique information it provides. Therefore, we present both mutability and standard deviation in a table.

### 2.3.3 Calculation method of highway load

In this study, traffic data indicators with a high correlation with transportation were selected to assess the highway load in
each province. The larger the corresponding highway load, the greater the economic losses when highway blocking events occur owing to high-impact weather conditions. Based on the statistical data issued by the National Bureau of Statistics, the data of mileage, passenger capacity, freight capacity, express capacity (the volume of express delivery), added value of transportation, and civilian automobile ownership were selected as the basic reference.

Since the area, altitude and topography in different regions affect the distribution of highways, and the degree of highway
utilization varies in different regions due to the differences in economic development, we perform a comprehensive assessment by referring to the highway density (highway length in a specific unit area and the highway load under the unit length).

The correlations between data items are higher because some of the data items are obtained by performing calculations from other data items. Additionally, subjective weight assignment methods, such as the gradation classification method and the
analytic hierarchy process, are not objective enough, while we prefer to obtain objective analysis from the data in this study. Highly correlated data were used to better fit the CRITIC method. The CRITIC method combines two dimensions–intensity of comparison and conflict–to combine the weights of the indicators. Comparative strength is expressed using the standard deviation, which reflects the degree of dispersion of data within the indicator, and conflict is expressed using the correlation coefficient, which reflects the correlation between indicators.

. Thus, the CRITIC weigh method is chosen. Specifically, data normalization is firstly performed for all traffic indicators, where $Z$ is the normalized data, calculated by $Z = \dfrac{Z_i - Z_{min}}{Z_{max} - Z_{min}}$ , and then the weights are assigned to each normalized data by using the CRITIC weight method to obtain the weight value of each indicator. So we develop an equation (Eq. (6)) to calculate the degree of highway load.

$$HW_{load} = H_d * \alpha + \Delta TF_{load} * \beta + GDP_{trans} * \gamma + \Delta P_{load} * \delta + VD * \varepsilon + EP * \epsilon + TF_{load} * \theta + P_{load} * \vartheta + V_{private} * \mu \quad (6),$$
where $HW_{load}$ denotes the highway load, $H_d$ the highway density, $\Delta TF_{load}$ the load capacity of freight transport for per kilometer, $GDP_{trans}$ the added value of the GDP generated by transportation, $\Delta P_{load}$ the number of people for per kilometer, VD the vehicle density, EP the number of express packages, $TF_{load}$ the total freight transport, $P_{load}$ the number of people, and $V_{private}$ the number of private vehicles. $\alpha$, $\beta$, $\gamma$, $\delta$, $\varepsilon$, $\epsilon$, $\theta$, $\vartheta$ and $\mu$ are the corresponding coefficient values of each parameter, these parameters will be computed according the above data, and these will be detailed calculated in the results part.

**3. Results**

**3.1 Highway blocking features**

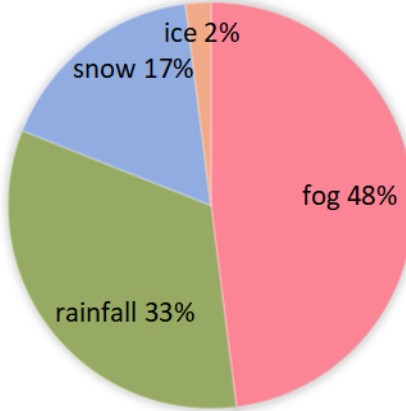

**Figure 2 main weather factors affecting highway blocking in China**

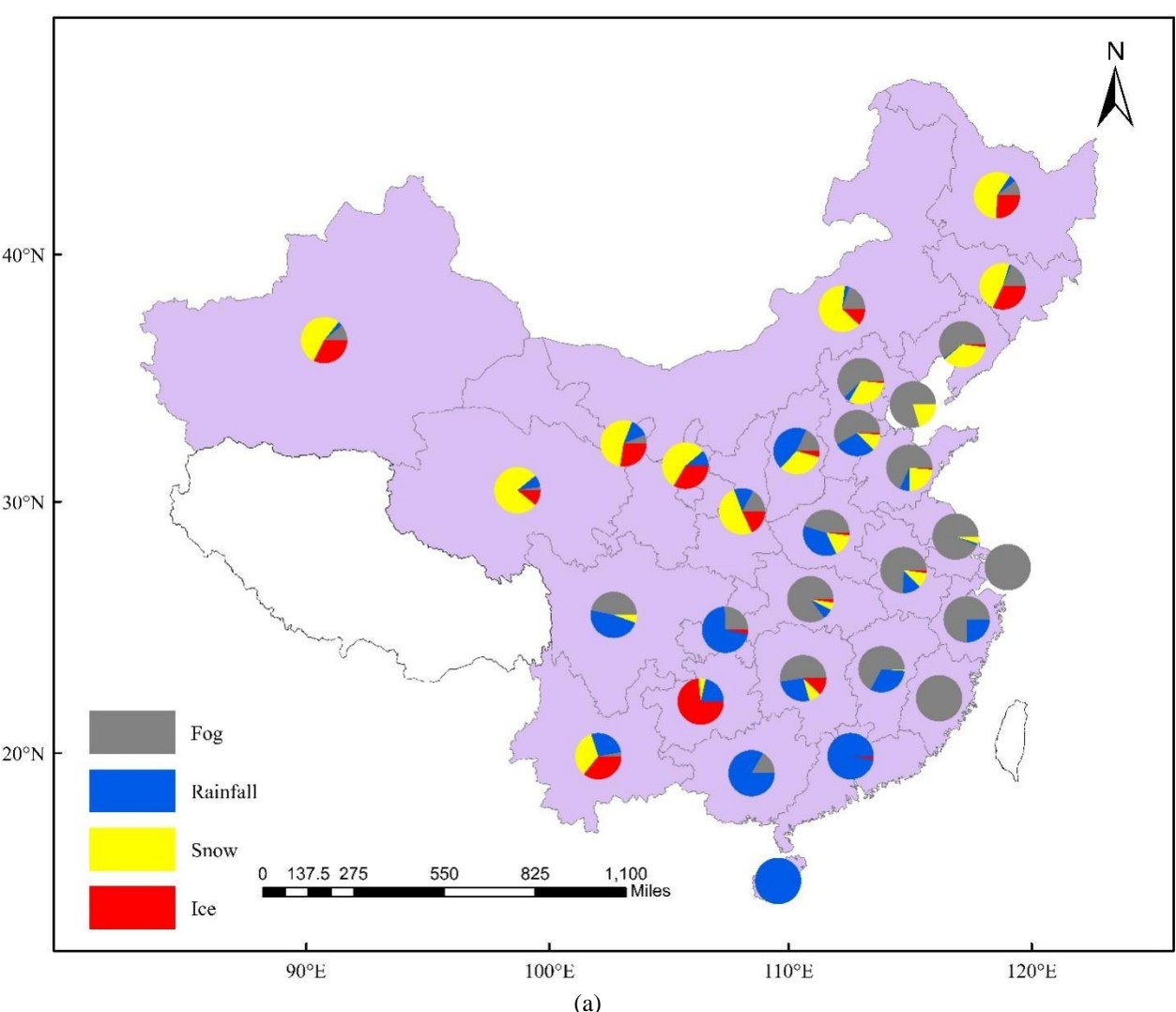

(a)

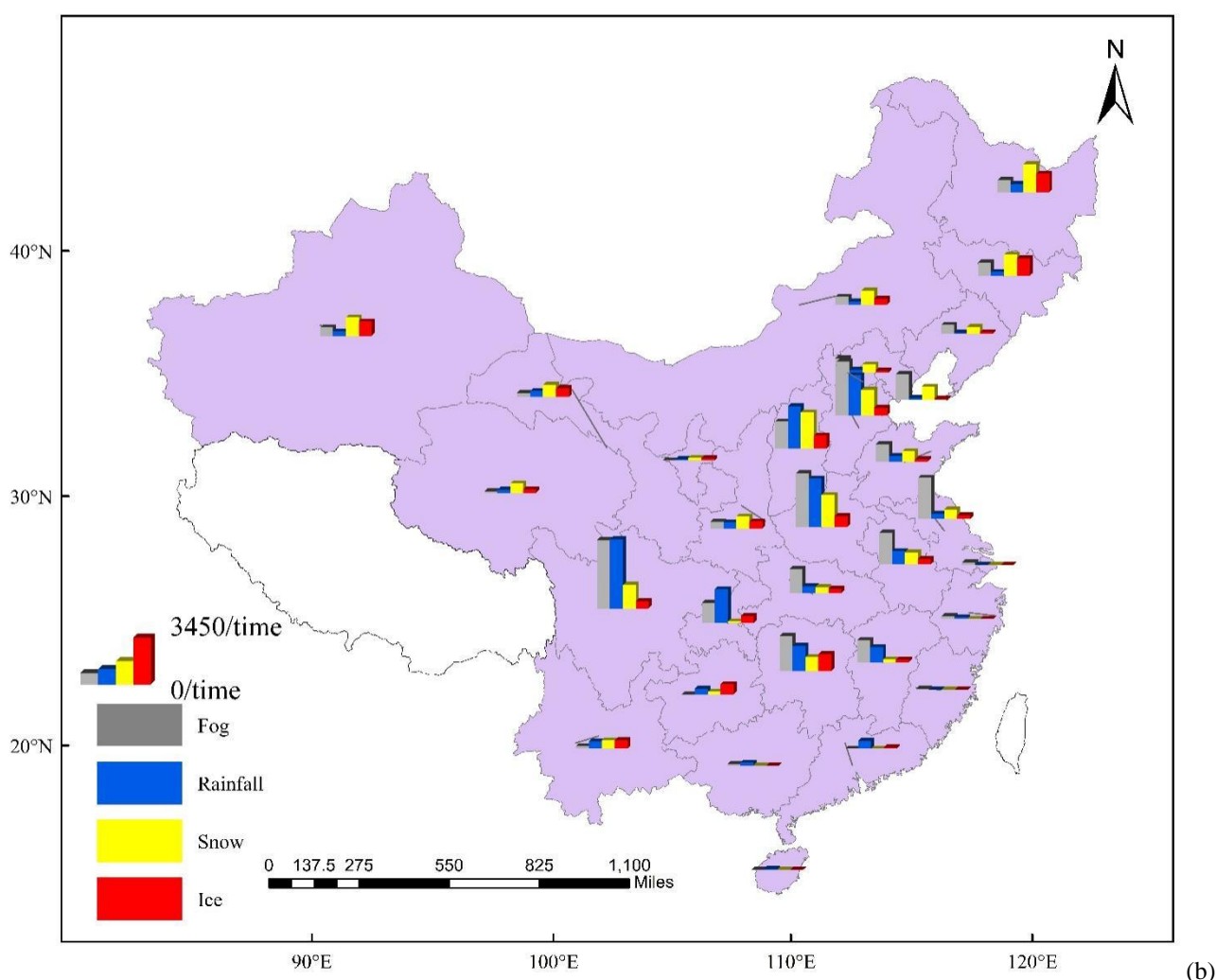

**Figure 3 High impact weather types leading to highway blocking in different provinces of China (a: proportion, b: the height of the bars represent number:)**

From figure 2 we can see that, fog is the main weather factor which causes highway blocking in China (Fig. 2), with a proportion of 48%. The next largest contributor is rainfall (road slippery), accounting for 33% of the total. The highway blocking caused by snowfall (snow cover) and icing also accounts for 17% and 2%, respectively. In addition, there are hail, gale, snowdrift (Wind blowing snow), typhoon, high temperature, dust and other weather factors, but the combined percentage was very low.

The main factors affecting highway blocking (Fig. 3a and Table 1) vary among provinces. In several provinces of Northeast China, Northwest China and the Qinghai-Tibet region, the main weather factors affecting highway blocking are snowfall and

icing, followed by dense fog. In several provinces of North China, East China and Central China, the main weather factor is dense fog, followed by snowfall or rainfall. The main weather factor in several provinces of South China is rainfall, followed by dense fog. The main weather factors in Yunnan and Guizhou Provinces of Southwest China are icing and snowfall, while it is dense fog and rainfall in Sichuan Province.

Although the main weather factors affecting highway blocking differ in different provinces of China (Fig. 3b and Table 1), the provinces with a higher frequency of highway blocking are Sichuan Province, Henan Province, Hebei Province, Shanxi Province, Hunan Province, Jiangsu Province and Chongqing Municipality. Rainfall in Sichuan, Chongqing, and Shanxi causes more blocking than dense fog. However, the blocking caused by dense fog is greater than that caused by rainfall in the Henan, Hebei, Hunan, and Jiangsu Provinces. Blocking events due to both dense fog and rainfall were much more frequent in Sichuan Province than in other provinces.

## 3.2 Spatio-temporal distribution features of highway blocking

There are large seasonal differences in highway blocking in various regions of China due to differences in geographical environment and climatic characteristics (Fig. 4a), and high-impact weather types (Fig. 4b), such as dense fog, snowfall (snow cover), rainfall (road slippery) and icing, etc. As shown in Fig. 4, the highway-blocking events are the most in Southwest China, reaching 8692 (29.6% of the total blocking events in China), followed by North China (8331). In Central China, the highway-blocking events also reach 7347.

The highway blocking in Southwest China is concentrated in summer and winter, with 3891 events (46.0%) and 3196 events (37.8%), respectively. The highway-blocking events in spring and autumn account for just 16.2%. The blocking events in Southwest China are mainly caused by rainfall (50.0%) and dense fog (42.5%), while those caused by snowfall and icing are much less, accounting for no more than 8% of the total. Summer rainfall in mountainous areas is heavy, often accompanied by thunderstorms and short-term heavy precipitation events. Mountain terrain is special, the speed of rainwater runoff is fast, easy to form flash floods or debris flow, these natural disasters are the main reasons for blocking the highway.

In North China, the highway-blocking events are also concentrated in summer and winter, with 2526 events (44.4%) and 2058 events (25.1%), respectively. The proportions of highway-blocking events in spring and autumn are 16.0% and 14.4%, respectively. The highway-blocking events in North China are mostly caused by dense fog (3760 events, 45.1%), followed by rainfall (2536 events, 30.4%). In addition, snowfall causes more blocking events in North China (1,833 events, 22.0%) than in Southwest China. Like North China, the highway-blocking events caused by dense fog and rainfall in Central China are the most frequent, i.e., 3757 events (51.1%) and 2366 events (32.2%), respectively.

There are 2283 highway-blocking events in East China, mainly concentrated in winter (43.5%) and spring (37.7%). Among them, 85.5% of blocking events are caused by dense fog, which is the highest proportion for all types resulting in blocking events in a single major region.

A total of 1828 highway-blocking events occurred in Northeast China, evenly distributed in spring, autumn, and winter, with a proportion of approximately 30% in these three seasons. Only 1268 blocking events occurred in Northwest China, mainly

in winter (nearly 50%). The highway-blocking events caused by snowfall accounted for nearly 50% in Northeast and Northwest China, followed by events caused by icing. The difference is that there are denser fog-caused blocking events in Northeast China than those in Northwest China.

The highway-blocking events in the Qinghai-Tibet region and South China are just 159. Among them, there are only 88 events in South China, most of which are concentrated in summer. However, the highway-blocking events in the Qinghai-Tibet region are concentrated in autumn and winter. The blocking events are mainly caused by snowfall in the Qinghai-Tibet region but rainfall in South China.

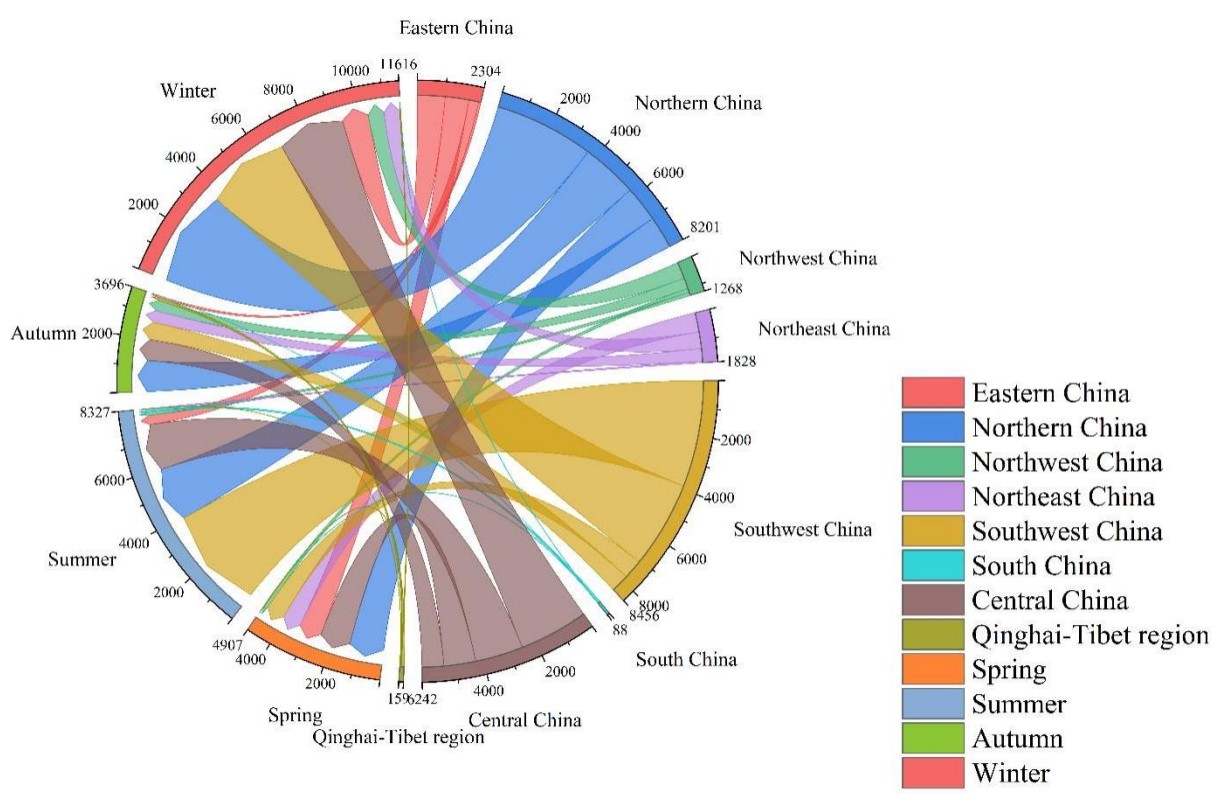

(a)

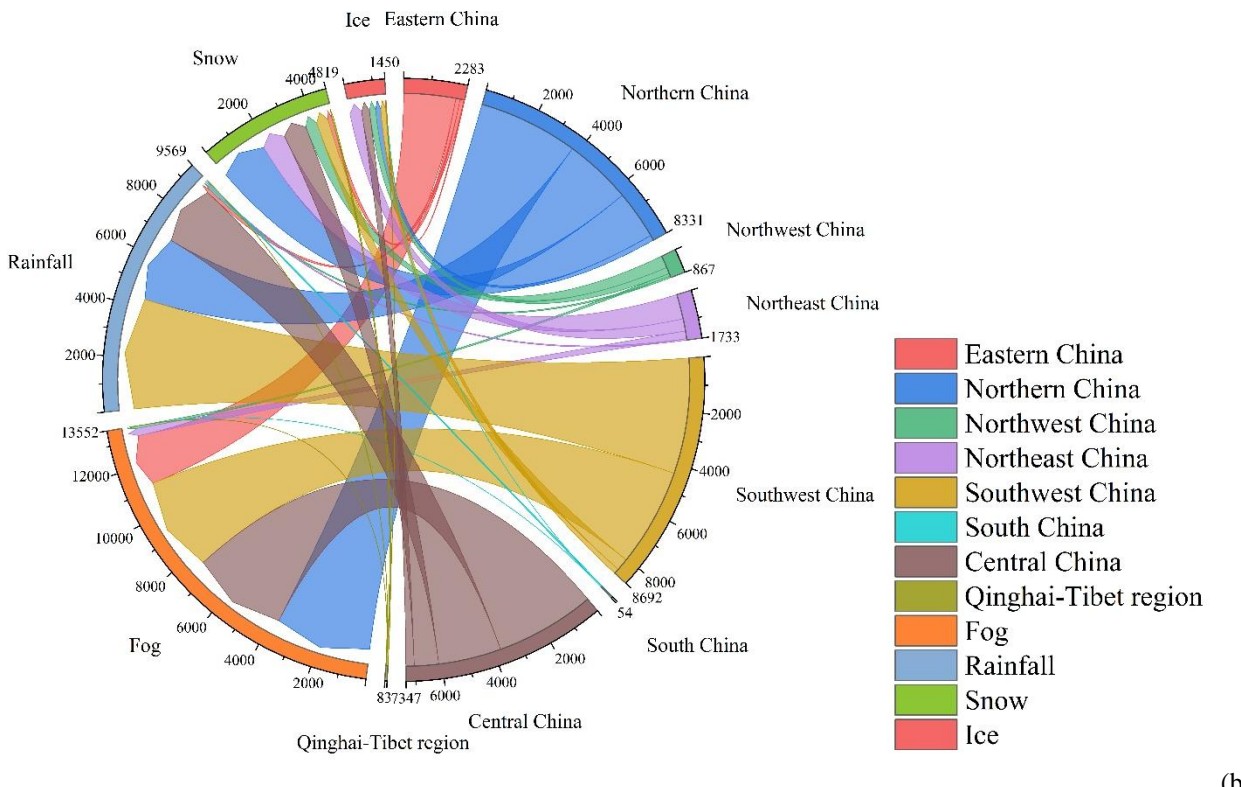

(b)

**Figure 4 Seasonal characteristics (a) and the main high impact weather (b) affecting the highway blocking in different areas of China**

From the ten-day distribution of highway-blocking events in different provinces of China (Fig. 5), it can be found that the regions where highway blocking occurs throughout the year are Chongqing and Sichuan of Southwest China and Henan, Hebei and Shanxi of North China. The peak of highway-blocking events differs in different provinces. The blocking events in Chongqing and Sichuan peak in late July, while those in the other provinces peak in November–December. The highway-blocking events in most provinces of China decrease noticeably from late August to early September. In Henan and Hebei Provinces, there are window periods in late March and middle to late September, which are two transition periods between winter and spring and between summer and autumn, with markedly reduced blocking events. In Sichuan and Chongqing, the window periods for blocking events are in late February and early March, especially in early March.

The diurnal variation of highway-blocking events in different provinces of China (Fig. 5b) suggests that they mostly occur from 17:00 BJT to 08:00 BJT in the following day, and fewer events appear at other time. Highway-blocking events appeared throughout the day in Chongqing, Xinjiang, Sichuan, Shanxi, Heilongjiang, Henan and Hebei, which is basically consistent with the ten-day variation result.

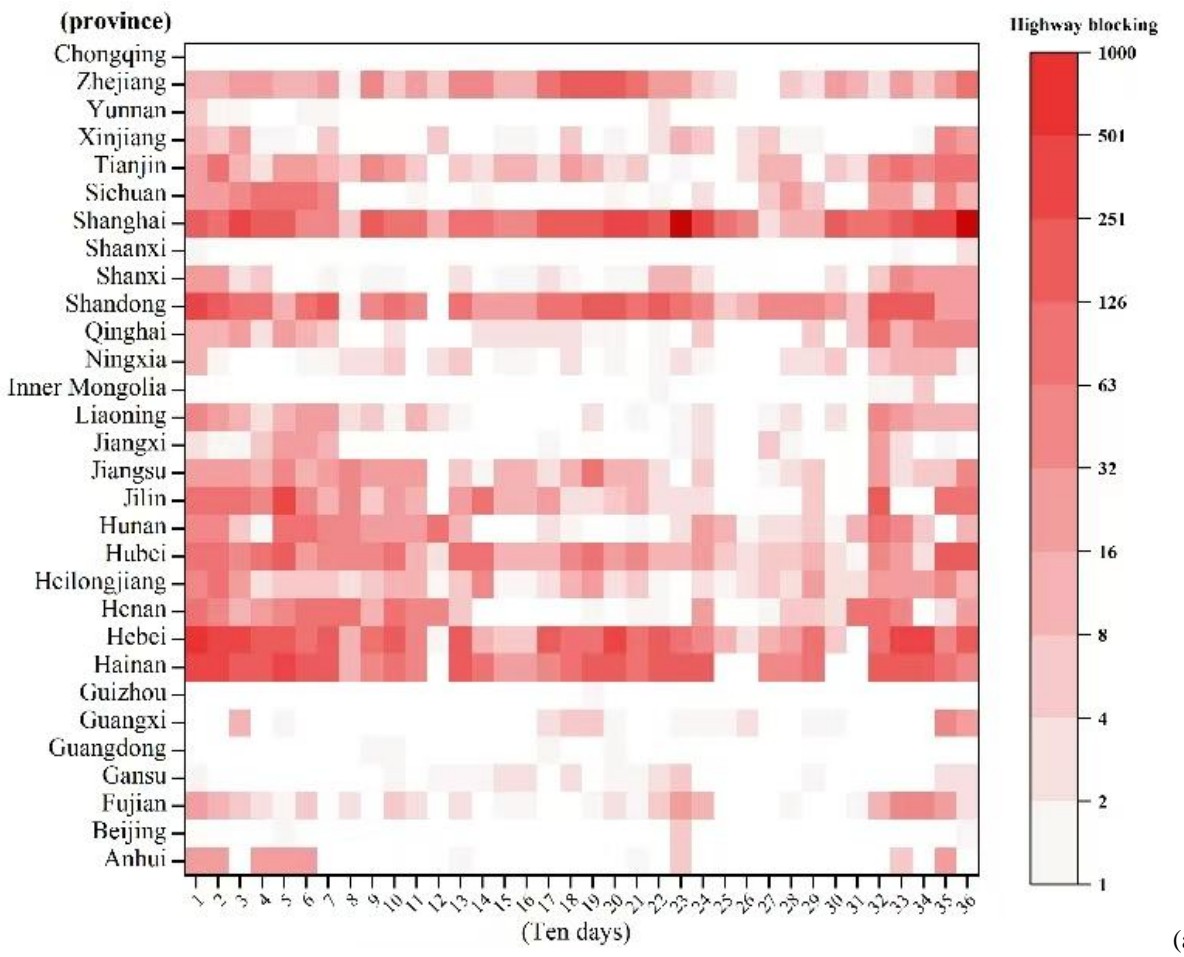

(a)

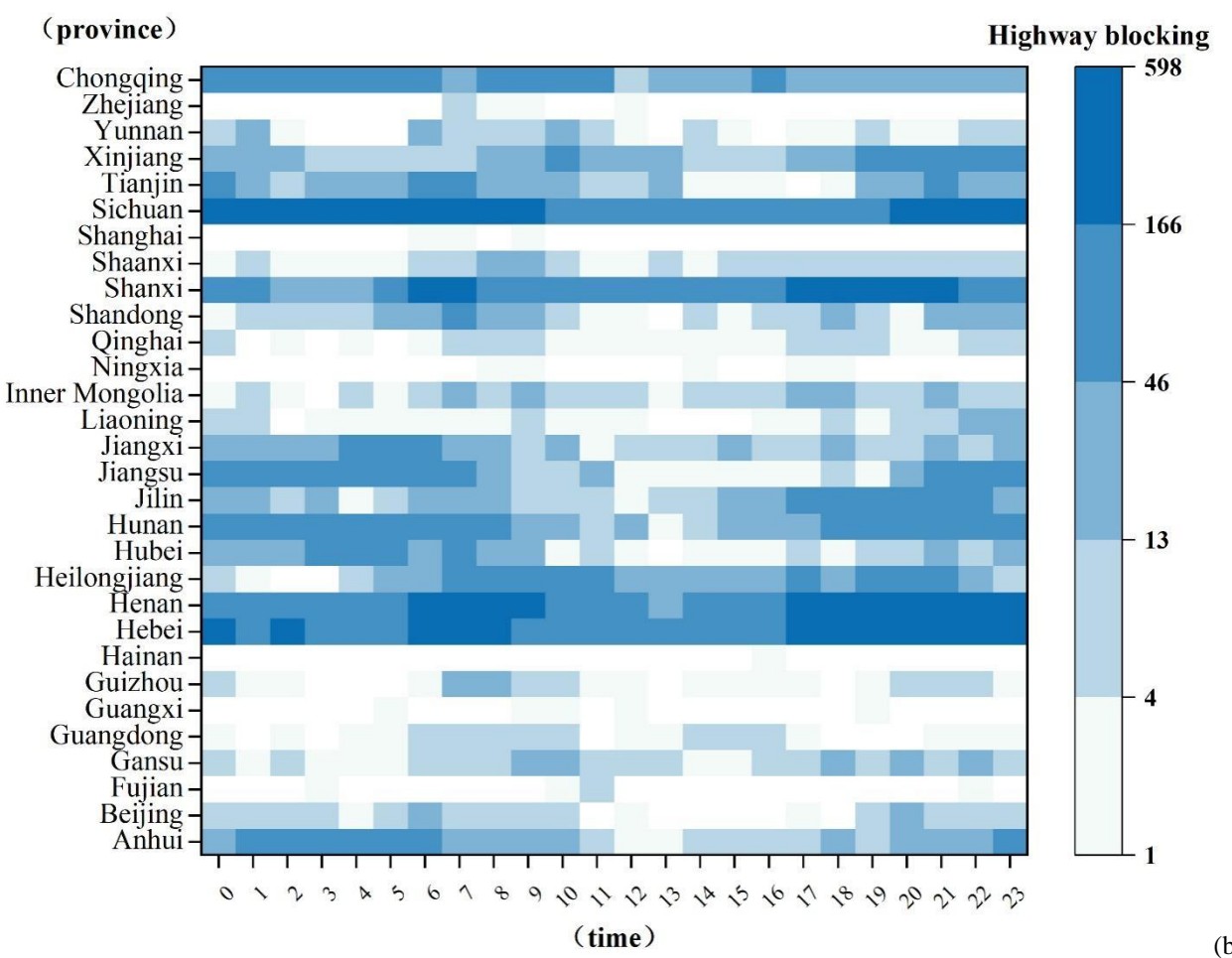

(b)

**Figure 5. The changing characteristics of highway blocking every ten days (a) and Diurnal variation (b)**

### 3.3 Highway blocking levels

To investigate the severity of highway blocking, we selected the blocking mileage (the distance of highway blocking),
blocking time, and response time as the most crucial reference indicators. This study only considered the evaluation of road
traffic by the blockage itself and did not consider the basic resources of the road network (the road miles per unit area, the
higher the density of road network per unit area, the more abundant road network resources) and the impact of secondary
disasters. If the road network resources are large, blocking may have little impact on the local road network, which is not
considered in the degree of blocking.

Firstly, the blocking mileage is used as the initial judgment condition of severity. Then, using equation 2, the blocking events
caused by different meteorological factors are clustered. The blocking mileage, blocking time and response time in different

high-impact weathers as the input vectors in the calculations. Finally, the severity of the blocking events is determined according to the size of the clustering centers.

Data normalization is performed for all severity levels of highway blocking due to meteorological factors. Then, the severity levels are judged according to the location and size of the cluster centers of the clustered mileage. The severity (S) of highway-blocking events is expressed as the equation:

$$S = L \times T \tag{7},$$

where $L$ indicates the blocking mileage (km), and $T$ denotes the blocking time (h). The severity of highway blocking is classified into five levels for clustering. The higher the level is, the more serious the blocking is. Furthermore, all highway-blocking events are classified according to the five clustering centers, and the distribution of the five levels of highway-blocking events is obtained (Table 2).

Table 3 shows the distribution of highway-blocking levels for the eight main regions in spring and summer. The regions with a severe level (level 4) and above in spring are mainly in Northeast China and North China, and the main high-impact weather for highway blocking is snowfall, fog, rainfall and icing in these areas. The blocking events with a moderate level can be found in most parts of China in spring, highly affected by snowfall, icing, dense fog and rainfall. The only blocking factors of dust and snowdrift appear in spring in Northwest China, while the blocking events in the Qinghai-Tibet region are caused by snowfall only. The levels in six regions of North China, Central China, South China, East China, Northwest and Southwest China all reach the severe level (level 4) in summer, and the main high-impact weather is rainfall. The moderate-level blocking events appear in most parts of China, mainly caused by rainfall and dense fog. In addition, there are blocking events caused by dust in Northwest China.

The distribution of highway-blocking levels in the eight major regions in autumn and winter (Table 4) indicates that the regions with blocking events at the severe level (level 4) and above in autumn are mainly in Northeast China, North China, Central China, Northwest China and the Qinghai-Tibet region, and these events are mainly influenced by snowfall, rainfall, fog and icing. The blocking events at the moderate level can be found in most parts of China and are also mainly caused by snowfall, rainfall, fog and icing. The most blocking events occur in winter. In this period, six regions of Northeast China, North China, Central China, South China, East China, Northwest China and Southwest China have the blocking events reaching the severe level (level 4), and the main high-impact weather is snowfall. The blocking events with a moderate level occur in most parts of China, mainly caused by dense fog, icing, snowfall and snowdrift. No blocking event occurs in South China throughout the winter. In contrast, the icing-caused blocking events increase in Northeast China and Central China. The snowdrift-caused blocking events and increased icing-caused blocking events can be observed in Northwest China. Compared with East China, no dense fog-caused blocking event appears in Northwest China.

Figure 6 presents the relationship between highway blocking levels and high-impact weather in China. Highway-blocking events are the highest frequency in Southwest China, and most of them are at slight (level 1) levels, of which 87.6% are at level 1. The blocking events with slight (level 1) and normal (level 2) levels are relatively fewer in Northern and Central

China; the blocking events at level 1 account for only 64.1% in North China and 62.6% in Central China. The percentages of blocking events at level 1 in Northwest and Northeast China are less than 50%.

The highway-blocking events at level 1 are the most, 50.3% of which are caused by dense fog. The percentages of blocking events caused by rainfall (road slippery) and snowfall (snow cover) reach 34.7% and 11.4%, respectively. Less than 5% of the blocking events at level 1 are caused by icing. In terms of the blocking events at lower levels (levels 1–3), the percentage of dense fog-caused blocking events increases with the severity level of blocking. For the blocking events at higher levels (levels 4–5), the proportion of blocking events caused by rainfall (road slippery) and snowfall (snow cover) is higher, especially at the highest level (level 5, exceeding 80%).

This result was analysed by a clustering algorithm, in which the clustering of blocking intensities showed a low distribution of high-level blocking events; usually disasters in which a specific long period of time and a wide range of meteorological hazards do not often occur in daily life. This is a presentation of the clustering results, sudden and prolonged disasters are not common, so the high level of highway blockage is less, in most cases, small-scale unexpected weather affects the smoothness of the traffic, but the highway management will also be able to deal with the problem as soon as possible, when it comes to the natural disasters of high intensity and wide range of the highway management can't unblock the traffic as soon as possible, therefore the intensity of the blockage events in this case will be very high, so the high level of the clustering results will be highlighted and present a scanty number of results.

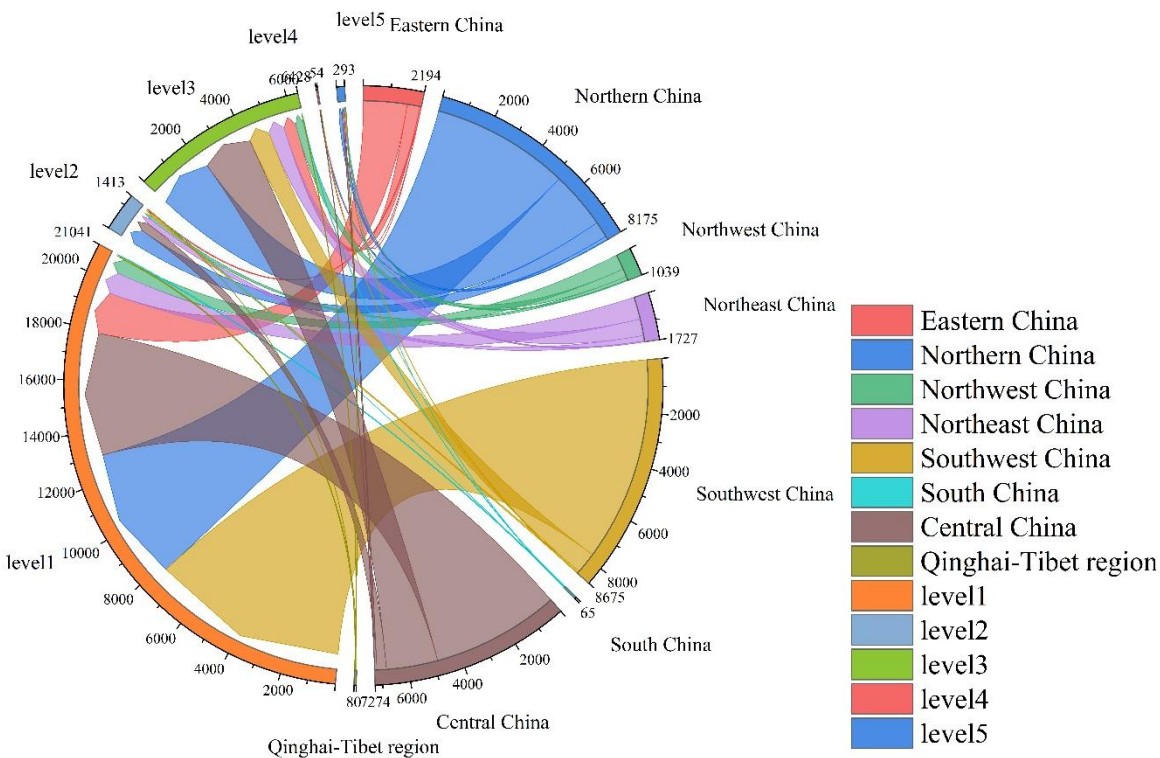

(a)

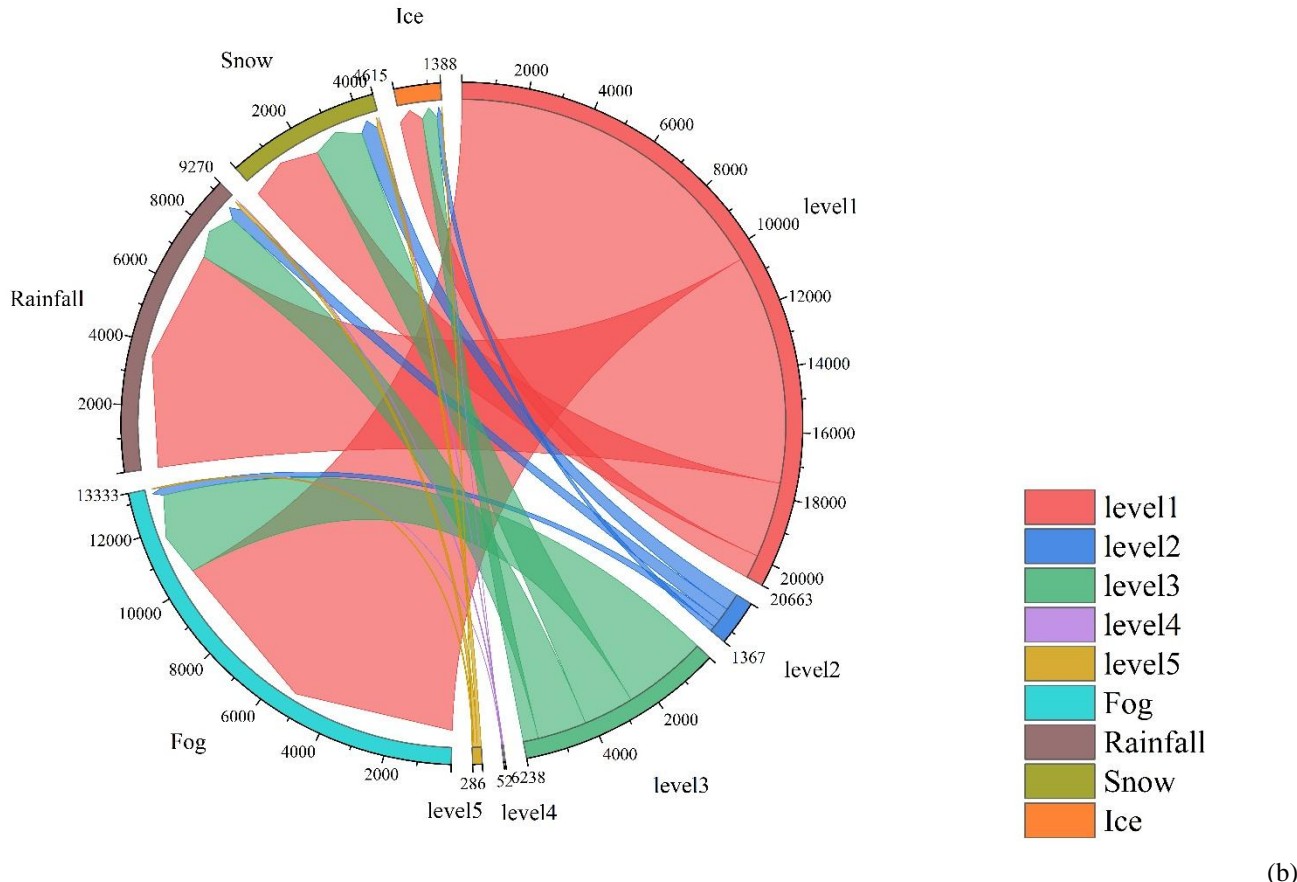

**Figure 6. The level of highway blocking (a) and relationship with high impact weather (b)**

### 3.4 Losses due to highway blocking

In this study, the economic indicators, $H_d$, $\Delta TF_{load}$, $GDP_{trans}$, $\Delta P_{load}$, VD and EP, are selected from the statistical data issued by the National Bureau of Statistics as the basic reference. Then, the economic volume per kilometer of $\Delta TF_{load}$, $GDP_{trans}$, $\Delta P_{load}$ and EP in different provinces of China is calculated by combining the highway mileages of different provinces (Table 5).

Next, the data normalization is performed for the selected economic indicator data. Furthermore, the CRITIC weigh method is used to assign weights for each normalized data, as shown in Table 6. The highway load in China calculated using Eq. (6) as below (Eq. (8)):

$$HW_{load} = H_d * 0.115 + \Delta TF_{load} * 0.1123 + GDP_{trans} * 0.1002 + \Delta P_{load} * 0.1174 + VD * 0.1079 + EP * 0.0984 +$$
$$TF_{load} * 0.1267 + P_{load} * 0.1190 + V_{private} * 0.1033 \tag{8},$$

The highway load of each province in China (Fig. 7) indicates that it shows an overall decreasing trend from the eastern coast to the inland. The high load is concentrated in Jiangsu, Shandong and Guangdong Provinces, and the highway load in

Jiangsu Province is the highest in China. In the inland provinces, Sichuan Province has a higher highway load than its neighbouring provinces due to its unique geographical location and economic development level in Southwest China. The overall highway load in Hainan Province is lower because of it is an island. Fujian Province has the lowest highway load among the coastal provinces.

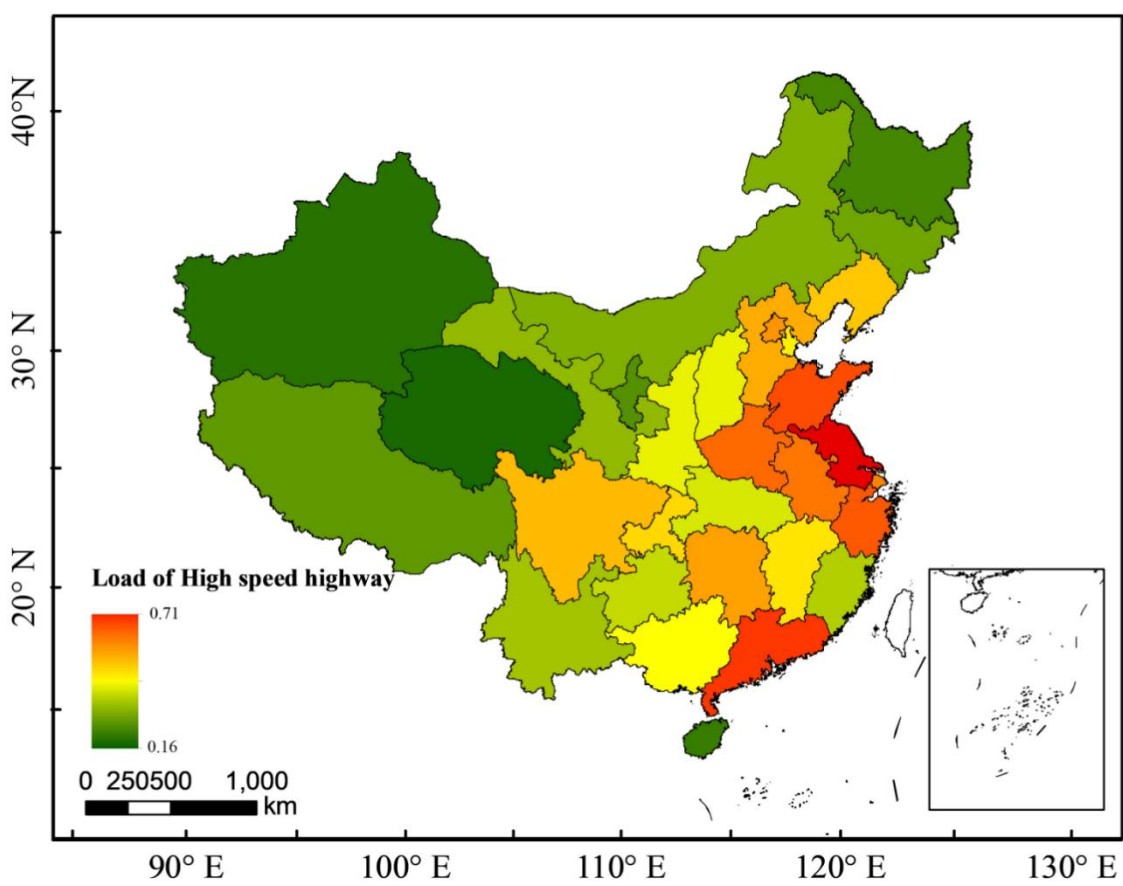

**Figure 7 the Highway Load in China**

**3.5 Economic Losses exposure evaluation due to fog-related highway blocking**

To assess the economic losses caused by highway blockages due to high-impact weather events, we collected data and established a model for economic loss exposure evaluation. We then use this model to calculate the economic losses caused by highway blockages resulting from high-impact weather events. Below, we compare specific high-impact weather events with the actual loss data to verify the accuracy and reliability of our assessment model.

Through the above analysis, we found that highway blocking caused by high-impact weather is related to weather and climate conditions in different regions and has regional and seasonal characteristics. However, as there are differences in the

economic indicators $H_d$, $\Delta TF_{load}$, $GDP_{trans}$, $\Delta P_{load}$, VD, and EP, there are significant differences in the highway load in different areas. When a highway-blocking event occurs, the higher the load, the greater are the losses and impacts. The definition of highway load proposed in sec 2.3.3 can provide a better assessment condition for understanding highway losses due to high-impact weather in each province. Using this new calculation method, we analysed and calculated all four types of high-impact weather. Fog is the main weather factor that causes highway blocking in China, and we selected the fog weather blocking loss distribution, which had the highest percentage, for a demonstration. In the following section, the impact of dense fog is assessed based on the load conditions in different provinces. The highway losses caused by each blocking event were calculated to obtain the $TF_{load}$, $GDP_{trans}$, $P_{load}$, and EP in the different provinces of China.

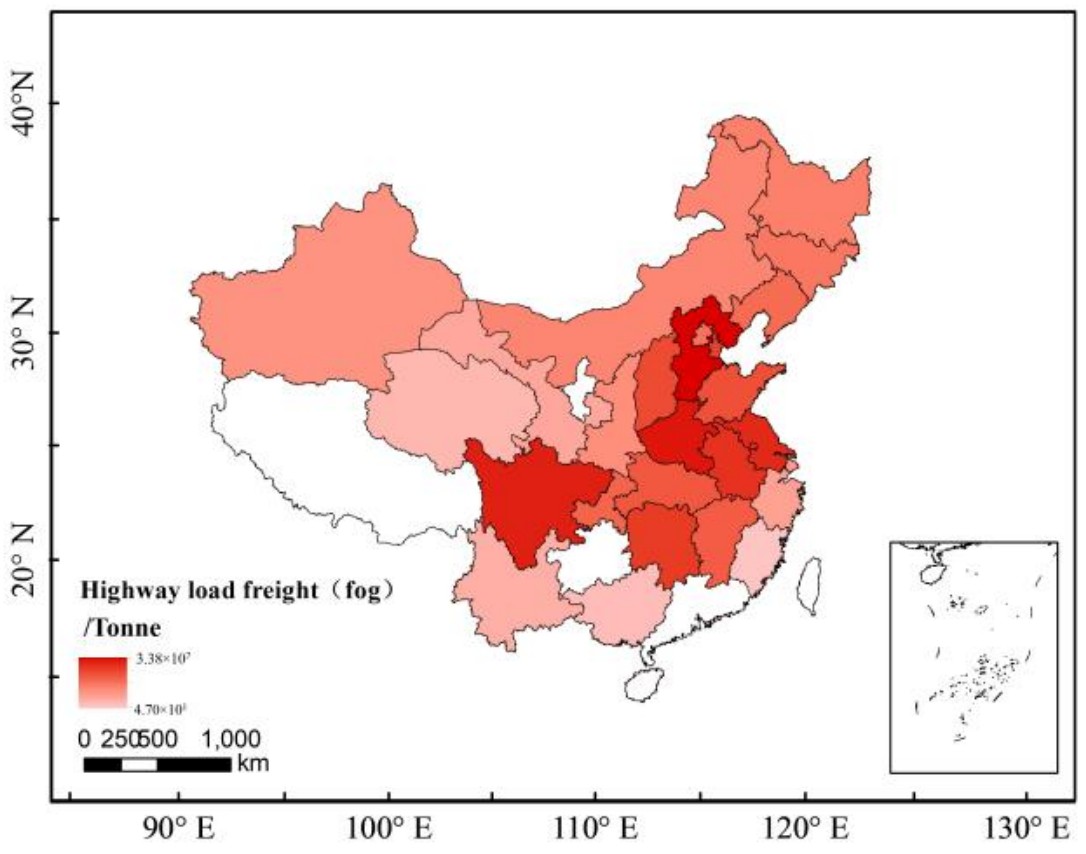

(a)

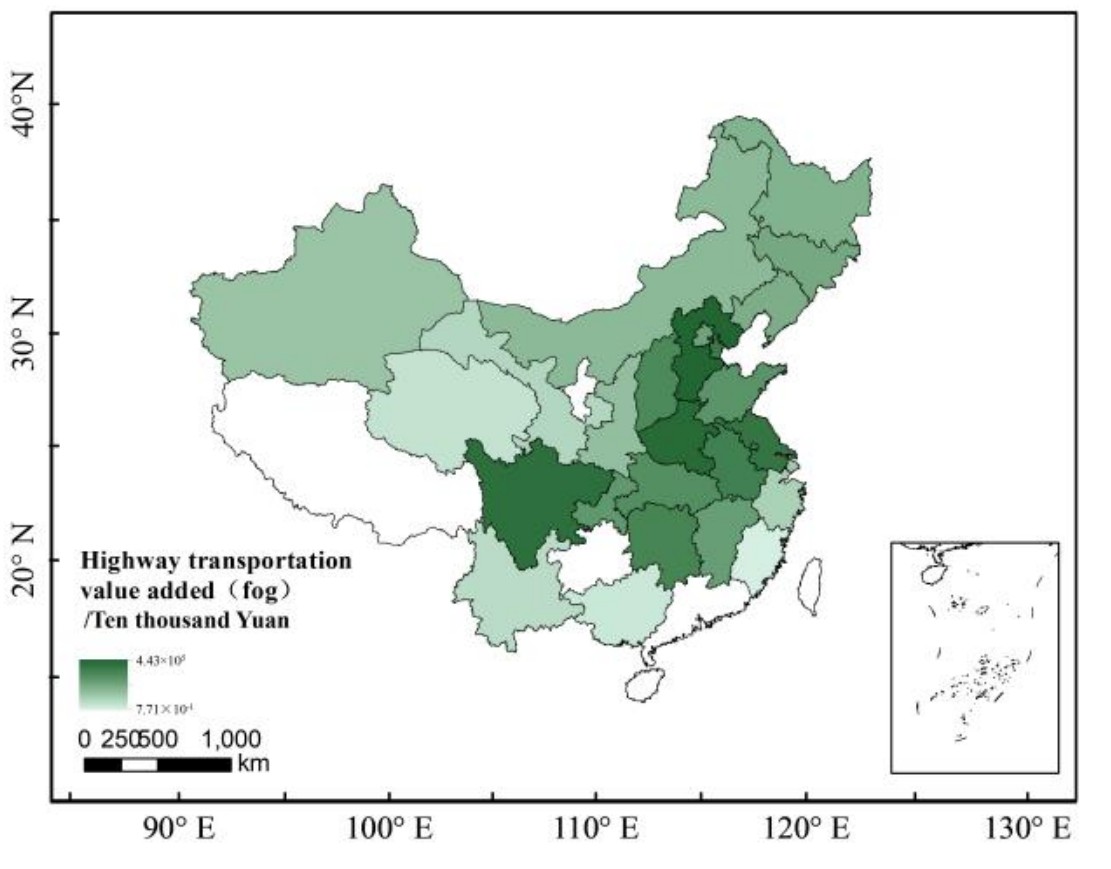

(b)

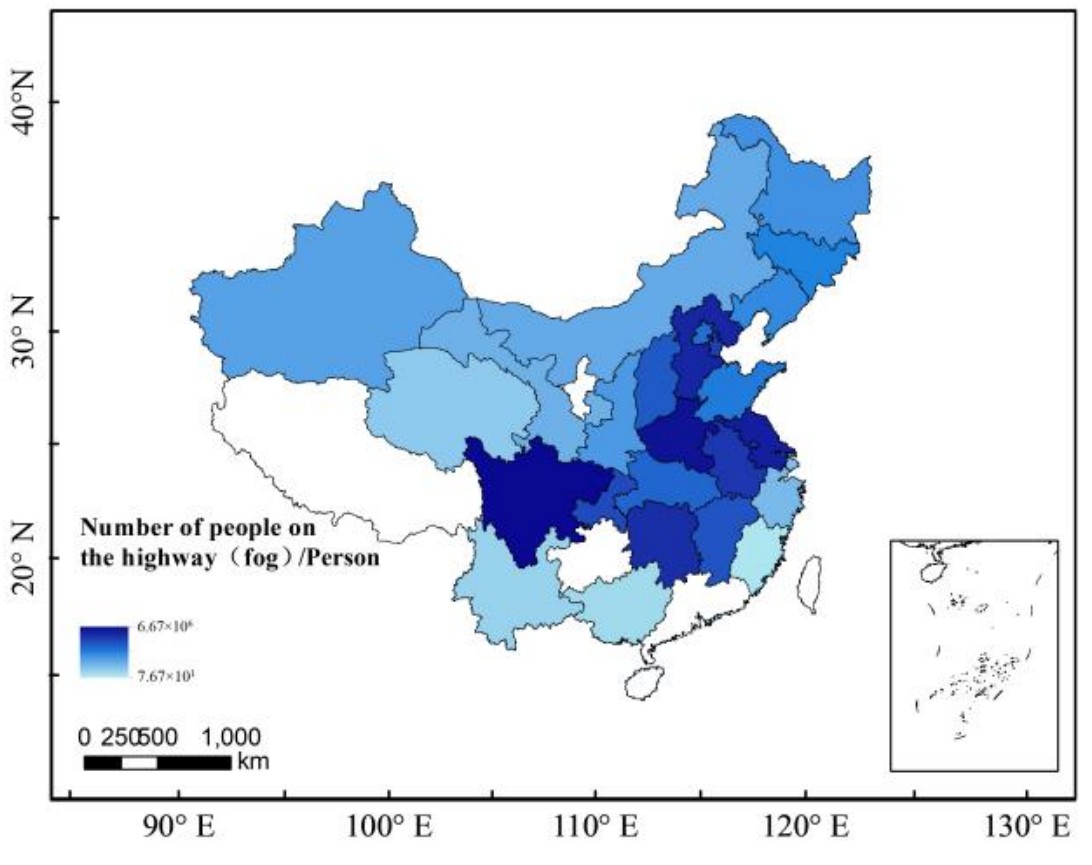

(c)

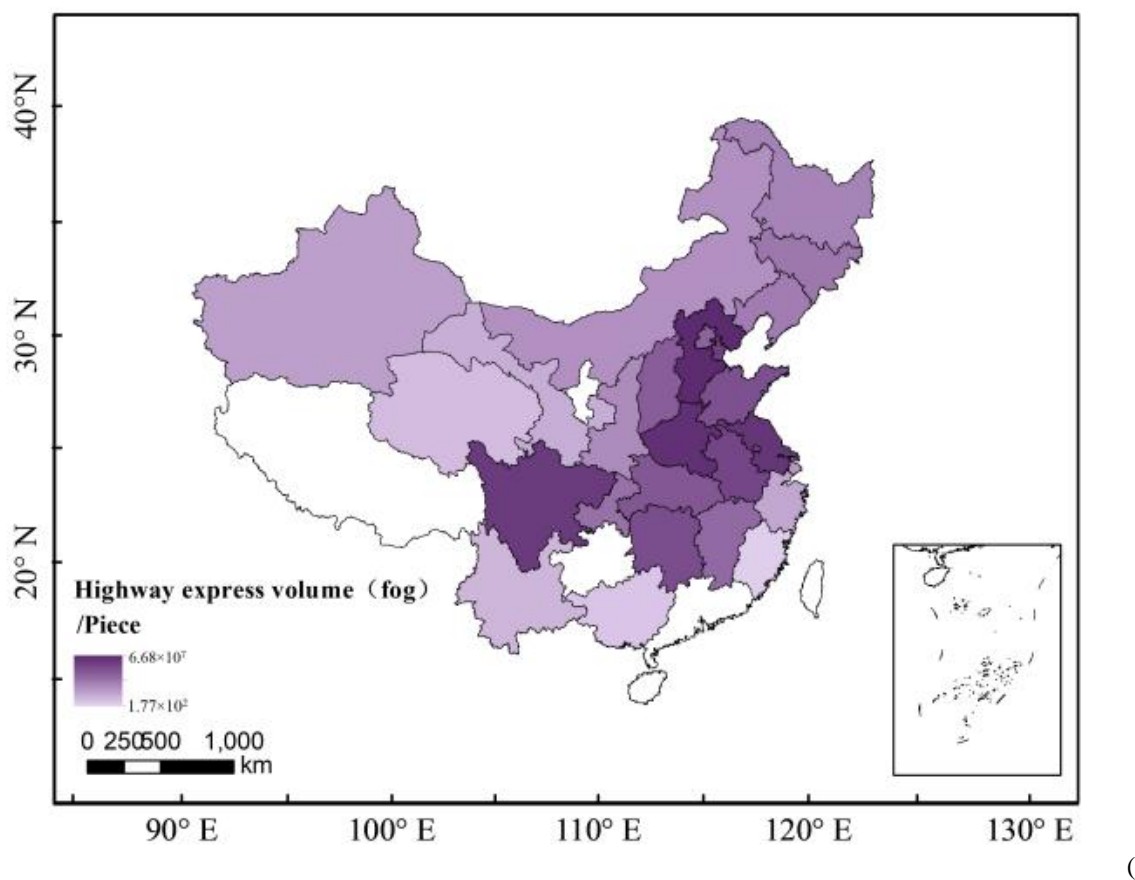

(d)

**Figure 8. Economic losses exposure evaluation caused by highway blocking due to fog (a: Delay or cancellation of freight transport; b: Transportation related GDP loss; c: Delay or impact on travel people; d: Delay or delay related express package) GDPtrans the added value of the GDP generated by transportation, EP the number of express packages, TFload the total freight transport, Pload the number of people.**

Figure 8 shows the highway losses exposure evaluation caused by dense fog. From the figures we can see that the highway losses caused by dense fog are mainly concentrated in North China, East China and Sichuan Province (especially in Jiangsu, Sichuan, Hebei and Henan Provinces), followed by Central China and Northeast China. However, the losses are remarkably low in Northwest China, South China and the Qinghai-Tibet region. No highway-blocking events caused by dense fog occur in Guangdong Province, Hainan Province, Guizhou Province and Tibet Autonomous Region throughout the year. From

figure 7 we can see that Guangdong Province have a high level of highway load, but have no highway-blocking events, this is because the main high-impact weather is rainfall (table 1).

Dense fog can cause highway blocking, making highway traffic delayed or affected. As shown in Table 7, due to dense fog, there are more than 1 million passengers delayed or affected in Hebei, Jiangsu, Anhui, Henan, Hunan and Sichuan Provinces and Tianjin Municipality. Three provinces, namely Jiangsu, Henan and Sichuan, are the most serious, with about 6.67

million, 6.39 million and 4.55 million passengers affected by dense fog weather in highway traffic.

The provinces with more serious delays in freight transportation caused by dense fog are Hebei, Henan, Jiangsu, Sichuan and Anhui, and the delays in freight transportation are more than 10 million tons. Freight transportation in three provinces, namely Hebei, Henan and Jiangsu, is affected the most, with 33.8 million, 26.7 million and 17.2 million tons, respectively. The express business in China is more developed, and most express packages are transported through highways. Therefore, the delays of express packages due to dense fog are also serious. More than ten million packages in Jiangsu Province, Hebei Province, Henan Province, Sichuan Province, Anhui Province and Tianjin City are delayed due to dense fog (Table 7), especially in Jiangsu (68.8 million packages), Hebei (59.0 million packages) and Henan (42.8 million packages). This result is basically consistent with the situation of freight transportation.

$GDP_{trans}$ is also affected by dense fog. There are 14 provinces with $GDP_{trans}$ losses of 100 million yuan or more (Table 7), especially in Hebei (4.43 billion yuan), Henan (3.93 billion yuan), Jiangsu (3.14 billion yuan), Sichuan (1.52 billion yuan) and Tianjin (1.37 billion yuan).

## 4. Discussions about the strengths and limitations of this methodology

Many scholars have conducted assessment studies on the economic losses caused by meteorological disasters. Due to the strong spatial distribution characteristics of meteorological disasters and the varying specific losses inflicted on different industries, the establishment of evaluation models tailored to specific industries and disasters, along with the calculated economic loss figures, are more conducive to providing scientific references for decision-makers. As provided in the current analysis, some scholars have also attempted to assess economic losses in disaster evaluation by adopting the multivariate linear regression-TOPSIS method to establish an evaluation model (Wang et al. 2023; Greema et al., 2020; Yu et al., 2020). By modeling the temporal evolution of 1.8 million trade relations between 7000 regional economic sectors, Kuhla et al (2021) studied the economic welfare loss from weather extreme events, their work found out that the regional responses to future extreme events are strongly heterogeneous in their resonance behavior.

Some scholars use the non-interoperable input-output model (IIM) to estimate the economic lost; They have emphasized the importance of providing knowledge on the most vulnerable areas from the point of view of causing disasters, as well as the importance of the economic losses of the most vulnerable areas. In addition, by introducing exposure and sensitivity as filtering processes, critical paths for interactions between components within the hazard system are constructed and practically applied. Overall, studies mentioned above are combined with already identified occurrence hazardous data and are aimed at loss studies in water resources, agriculture, and economy, ignoring the assessment of losses in transportation, and a distinction between hazard categories (Bhattacharyya et al., 2021; Khalid et al., 2020; Dhunny et al., 2020). Therefore, this paper attempts to use the existing disaster record data and combine it with economical data to make a loss assessment analysis.

During the implementation process, following points need to be evaluated carefully and found to affect the results significantly: 1. Ensure the quality and accuracy of the data to obtain reliable assessment results. 2. Consider the impact of

different regions and various types of high-impact weather events on highway blockages, in order to develop more targeted management strategies. 3. Regularly update the assessment model to adapt to the continuously changing environment and societal needs. Through the newly established method, we can better evaluate the economic losses caused by highway blockages due to high-impact weather events, thereby providing strong support for highway management and control.

Few studies have assessed disasters caused by high-impact weather on highway traffic losses. In the current analysis, the relationship between high-impact weather and economic losses on highways was studied through data mining. The weight coefficients representing the extreme weather events were derived entirely from high-impact weather and economic data without any human intervention, and a strong correlation was found between the variables. Owing to the large geographical area covered by the data and the short time series, the observed differences vary across provinces. We hope that future research will further explore the sensitivity of severe weather indicators.

Through the analysis of Economic Losses due to Fog-Related Highway Blocking in Section 3.5, we found that the differences between Figures 7 and 8 mainly stem from variations in economic indicators within each province, leading to different economic loss values. Thus, the sensitivity of the outcome responds well to parameter variations. Additionally, different provinces experience high-impact weather in different seasons, resulting in corresponding variations in economic indicators and economic losses. Therefore, the equation (Eq. 8) is directly related to the various economic indicators of the corresponding provinces as well as the seasons in which high-impact weather occurs.

Overall, our analysis based on Eq. 8 can effectively estimate and evaluate the economic losses caused by highway congestion owing to high-impact weather events. However, the limitation of our approach is that the lack of economic data in different regions can lead to bias in the results. Therefore, the evaluation model in the present study needs to be updated regularly to adapt to changing environmental and social conditions.

## 5. Conclusions

High-impact weather events often lead to highway blocking, which in turn causes serious economic and human losses. This study proposes a method for assessing the economic loss exposure evaluation caused by high-impact weather events that lead to highway blockages to facilitate the management and control of highways and the evaluation of economic losses.

Based on the K-means cluster analysis and CRITIC weight assignment method, we analysed the highway blocking and highway load features for each province based on economic losses. High-impact weather events include fog, rainfall, snowfall (snow cover), icing, hail, gale, snowdrift (wind blowing snow), typhoons, high temperatures, and dust.

The following conclusions are obtained from this study:

- A new method developed to assess the extreme weather impact on highway load and related to economical losses.

- The overall seasonal distribution of highway-blocking events in China displays a pattern of more in winter and summer, and less in spring and autumn.

- The highway-blocking events are characterized by diurnal variation, and that appear from 17:00 BJST to 08:00 BJST the following day.

- When a highway-blocking event occurs, the higher the load is, the greater the losses assessment and impacts are. The highway load shows a decreasing trend from the eastern coast to the inland. Specifically, the high load is mainly concentrated in three provinces of Jiangsu, Shandong and Guangdong, with the highest in Jiangsu Province. Guangdong Province, as an economically developed province in China that has many harbors.

- The economic losses resulting from highway-blocking events caused by dense fog is found one of the important factors for economic losses. The results indicate that the highway losses caused by dense fog are mainly concentrated in Northern China, Eastern China and Southwestern China. Hebei, Henan, Jiangsu, Sichuan, and Tianjin suffer the most $GDP_{trans}$ losses at 4.43 billion (due to fog), 3.93 billion (due to fog), 3.14 billion (due to fog), 1.52 billion (due to fog) and 1.37 billion (due to fog) yuan, respectively.

- The highway-blocking data used in this study is only for the year 2020 as a test year, and additional time series of observations are needed to further validate the results. The assessment of losses is only judged by the degree of highway load but economic models can be employed to continue refining the research results.

The research presented in this paper brings up the need to further research is necessary on the impact of high-impact weather events on highway blocking. This will enhance our ability to assess economic losses caused by such weather conditions and improve the resilience and response capabilities of highways to meteorological disasters. Additionally, it is crucial to develop advanced predictive models and early warning systems to mitigate the adverse effects of high-impact weather, thereby ensuring the safety and efficiency of highway transportation networks.

**Author contribution:**

DL and MY designed the experiments and TJ carried them out; YB, HW, and FZ performed the measurements; DL, MY, and TJ analysed the data; DL and TJ wrote the manuscript draft; DL, TJ, MY, IG and YB reviewed and edited the manuscript.

**Competing interests**

The authors declare that they have no conflict of interest.

**Acknowledgement**

This work was jointly supported by the National Natural Science Foundation of Jiangsu, China (BK20231396), the 333 Project of Jiangsu Province, the Beijige Open Foundation (BJG202201).

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

**Tables**

**Table 1 the main weather factors affecting highway blocking in different provinces, China**

**\*,\*\*, no data.**

| area | province | High impact weather types | | | | |
|---|---|---|---|---|---|---|
| | | Fog | Rainfall | Snow | Icing | total |
| North China | Beijing | 90 | 6 | 46 | 2 | 144 |
| | Hebei | 2376 | 1184 | 479 | 43 | 4082 |
| | Shanxi | 534 | 1313 | 957 | 125 | 2929 |
| | Inner Mongolia | 45 | 7 | 147 | 27 | 226 |
| | Shandong | 228 | 23 | 79 | 4 | 334 |
| | Tianjin | 487 | 3 | 125 | 1 | 616 |
| Northeast China | Heilongjiang | 103 | 46 | 571 | 244 | 964 |
| | Jilin | 126 | 10 | 321 | 214 | 671 |
| | Liaoning | 59 | 1 | 36 | 2 | 98 |
| Northwest China | Shaanxi | 35 | 28 | 105 | 37 | 205 |
| | Gasnsu | 12 | 25 | 104 | 53 | 194 |
| | Ningxia | 0 | 1 | 5 | 3 | 9 |
| | Xinjiang | 51 | 14 | 245 | 149 | 459 |
| Qinghai-Tibet Region of China | Qinghai | 2 | 7 | 65 | 9 | 83 |
| | Tibet* | 0 | 0 | 0 | 0 | 0 |
| East China | Shanghai | 5 | 0 | 0 | 0 | 5 |
| | Jiangsu | 1238 | 19 | 64 | 10 | 1331 |
| | Anhui | 703 | 119 | 98 | 19 | 939 |
| | Zhejiang | 6 | 2 | 0 | 0 | 8 |
| Central China | Hubei | 409 | 34 | 25 | 11 | 479 |
| | Jiangxi | 352 | 166 | 6 | 5 | 529 |
| | Hunan | 905 | 464 | 144 | 212 | 1725 |
| | Henan | 2091 | 1702 | 738 | 83 | 4614 |
| South China | Guangdong | 0 | 45 | 0 | 1 | 46 |
| | Guangxi | 1 | 5 | 0 | 0 | 6 |
| | Fujian | 1 | 0 | 0 | 0 | 1 |
| | Taiwan** | 0 | 0 | 0 | 0 | 0 |
| | Hainan | 0 | 1 | 0 | 0 | 1 |
| Southwest China | Chongqing | 296 | 834 | 2 | 36 | 1168 |
| | Sichuan | 3393 | 3450 | 404 | 35 | 7282 |
| | Guizhou | 0 | 22 | 5 | 75 | 102 |
| | Yunnan | 4 | 38 | 48 | 50 | 140 |

**Table 2 clustering center of the highway blocking in China**

|  | Level | clustering center(km*h) |
|---|---|---|
| 1 | slight | 143.37 |
| 2 | mild | 870.41 |
| 3 | moderate | 2019.75 |
| 4 | severe | 3933.28 |
| 5 | extreme | 6610.59 |

**Table 3 The main weather factors affecting highway blocking and the levels in Spring and Summer in China**

|  | area | level1 | level2 | level3 | level4 | level5 |
|---|---|---|---|---|---|---|
| Spring | Northeast China | snow, icing, fog | snow, icing | snow, icing, fog | snow, fog | snow, icing |
|  | North China | fog, rainfall, snow | rainfall, fog, snow | rainfall, fog, snow | snow | rainfall, snow |
|  | Central China | fog, rainfall | rainfall | rainfall, fog | / | rainfall |
|  | South China | rainfall | / | rainfall | / | / |
|  | East China | fog | / | fog | / | / |
|  | Southwest China | rainfall, fog | snow | snow, fog | / | / |
|  | Northwest China | dust, snow | dust, snow | dust, snowdrift, snow | / | / |
|  | Qinghai-Tibet Region of China | snow | / | snow | / | / |
| Summer | Northeast China | rainfall, fog | / | fog, rainfall | / | / |
|  | North China | rainfall, fog | rainfall, fog | rainfall, fog | rainfall | rainfall, fog |
|  | Central China | rainfall, fog | rainfall | rainfall, fog | rainfall | rainfall |
|  | South China | rainfall | rainfall | rainfall | rainfall | rainfall |
|  | East China | fog, rainfall | rainfall | rainfall, fog | rainfall | rainfall |
|  | Southwest China | rainfall, fog | rainfall | rainfall | rainfall | rainfall |
|  | Northwest China | dust, rainfall | dust | dust, rainfall | rainfall | rainfall |
|  | Qinghai-Tibet Region of China | rainfall | / | rainfall | / | / |

/, no data.

**Table 4 The main weather factors affecting highway blocking and the levels in Autumn and Winter in China**

| | area | level1 | level2 | level3 | level4 | level5 |
|---|---|---|---|---|---|---|
| Autumn | Northeast China | snow, icing, fog | icing, snow | icing, snow, fog | / | snow, icing |
| | North China | fog, rainfall, snow | snow, rainfall, fog | fog, rainfall, snow | snow | snow, fog, rainfall |
| | Central China | fog, rainfall, snow | rainfall | rainfall, fog, snow | rainfall | rainfall |
| | South China | typhoon, rainfall | / | / | / | / |
| | East China | fog | / | fog | / | / |
| | Southwest China | fog, rainfall | / | rainfall, fog | / | / |
| | Northwest China | icing, snow, fog | snow | snow, fog | / | snow |
| | Qinghai-Tibet Region of China | snow | snow | / | / | / |
| Winter | Northeast China | snow, fog, icing | snow, icing | snow, icing, fog | / | snow |
| | North China | fog, snow | snow, fog | fog, snow | snow | snow, fog |
| | Central China | fog, snow, icing | snow, fog, icing | fog, snow, icing | / | snow |
| | South China | / | / | / | / | / |
| | East China | fog, snow | snow, fog | fog, snow | / | / |
| | Southwest China | fog, snow | snow, icing | fog, snow | / | / |
| | Northwest China | snow, icing, fog | snow, snowdrift, icing | snow, icing, snowdrift | / | snow, icing |
| | Qinghai-Tibet Region of China | snow | / | snow, icing | / | snow, icing |

/, no data.

**Table 5 Highway length, freight transport for per km, GDP generated by transportation, people for per Km, express package for per Km.**

| | Length | $\Delta TF_{load}$ | $GDP_{trans}$ | $\Delta P_{load}$ | EP |
|---|---|---|---|---|---|
| | km | 10,000-ton/km | Ten thousand Yuan/km | 10thousand/km | 10,000 pieces/km |
| Beijing | 1200 | 18.16 | $7.0\times10^3$ | 20.46 | 198.52 |
| Tianjin | 1300 | 24.82 | $6.2\times10^3$ | 6.10 | 71.36 |
| Hebei | 7800 | 27.17 | $3.6\times10^3$ | 1.36 | 47.47 |
| Shanxi | 5700 | 17.23 | $1.8\times10^3$ | 1.31 | 9.40 |
| Inner Mongolia | 7000 | 15.57 | $1.6\times10^3$ | 0.46 | 2.79 |
| Liaoning | 4300 | 32.23 | $2.8\times10^3$ | 6.10 | 26.04 |
| Jilin | 4300 | 8.90 | $1.4\times10^3$ | 2.66 | 10.39 |
| Heilongjiang | 4500 | 7.89 | $1.1\times10^3$ | 1.69 | 10.12 |
| Shanghai | 800 | 57.56 | $1.94\times10^4$ | 1.67 | 420.41 |
| Jiangsu | 4900 | 35.64 | $6.5\times10^3$ | 13.81 | 142.38 |
| Zhejiang | 5100 | 37.17 | $3.8\times10^3$ | 7.62 | 351.89 |
| Anhui | 4900 | 49.70 | $3.8\times10^3$ | 4.65 | 44.94 |
| Fujian | 5600 | 16.27 | $2.7\times10^3$ | 2.66 | 61.28 |
| Jiangxi | 6200 | 22.89 | $1.7\times10^3$ | 5.43 | 18.07 |
| Shandong | 7500 | 35.63 | $4.6\times10^3$ | 2.60 | 55.36 |
| Henan | 7100 | 27.27 | $4.0\times10^3$ | 6.52 | 43.66 |
| Hubei | 7200 | 15.88 | $2.4\times10^3$ | 3.02 | 24.79 |
| Hunan | 7000 | 25.21 | $2.1\times10^3$ | 6.31 | 21.02 |
| Guangdong | 10500 | 22.02 | $3.2\times10^3$ | 5.23 | 210.30 |
| Guangxi | 6800 | 21.37 | $1.3\times10^3$ | 3.94 | 11.45 |
| Hainan | 1300 | 5.27 | $2.0\times10^3$ | 3.51 | 8.47 |
| Taiwan* | 0 | 0 | 0 | 0 | 0 |
| Chongqing | 3400 | 29.32 | $2.8\times10^3$ | 9.25 | 21.50 |
| Sichuan | 8100 | 19.46 | $1.7\times10^3$ | 5.59 | 26.56 |
| Guizhou | 7600 | 10.45 | $9.0\times10^2$ | 4.42 | 3.70 |
| Yunnan | 8400 | 13.76 | $1.2\times10^3$ | 2.29 | 7.50 |
| Tibet** | 100 | 40.39 | $4.4\times10^3$ | 5.76 | 11.39 |
| Shaanxi | 6200 | 18.72 | $1.8\times10^3$ | 4.77 | 14.80 |
| Gansu | 5100 | 12.01 | $8.0\times10^2$ | 4.41 | 2.71 |
| Qinghai | 3500 | 3.10 | $3\times10^2$ | 0.95 | 0.67 |
| Ningxia | 1900 | 18.01 | $1.0\times10^3$ | 1.53 | 3.85 |
| Xinjiang | 5600 | 7.20 | $1.1\times10^3$ | 0.88 | 2.05 |

*,**, no data.

**Table 6 Weight analysis of different economic indicators**

| items | Indicators variability | Indicators conflict | Information content | average | Standard deviation | weight |
|---|---|---|---|---|---|---|
| MMS_highway density (km/km2) | 0.22 | 5.403 | 1.19 | 0.295 | 0.220 | 0.1150 |
| MMS_the volume of freight transport per km (10,000-ton/km) | 0.235 | 4.948 | 1.162 | 0.356 | 0.235 | 0.1123 |
| MMS_$GDP_{trans}$ (100 million yuan/) | 0.278 | 3.734 | 1.037 | 0.368 | 0.278 | 0.1002 |
| MMS_$P_{load}$ per km (10thousand /km) | 0.203 | 5.984 | 1.215 | 0.214 | 0.203 | 0.1174 |
| MMS_vehicle density(10thousand /km) | 0.201 | 5.56 | 1.117 | 0.134 | 0.201 | 0.1079 |
| MMS_package volume (100 million) | 0.223 | 4.57 | 1.018 | 0.121 | 0.223 | 0.0984 |
| MMS_ the volume of freight transport (10,000-ton) | 0.286 | 4.587 | 1.311 | 0.405 | 0.286 | 0.1267 |
| MMS_thevolumeof road passenger (10thousand) | 0.258 | 4.765 | 1.231 | 0.323 | 0.258 | 0.1190 |
| MMS_civilian car ownership (10thousand) | 0.268 | 3.993 | 1.069 | 0.331 | 0.268 | 0.1033 |

615

**Table 7 Personnel travel delay, freight delay, GDP generated by transportation, express delivery volume in fog weather**

| Province | $P_{load}$ person | $TF_{load}$ tonne | $total-GDP_{trans}$ Ten thousand Yuan | $EP$ piece |
|---|---|---|---|---|
| Beijing | $8.76 \times 10^5$ | $7.78 \times 10^5$ | $3.01 \times 10^4$ | $8.50 \times 10^6$ |
| Tianjin | $1.49 \times 10^6$ | $6.07 \times 10^6$ | $1.52 \times 10^5$ | $1.75 \times 10^7$ |
| Hebei | $1.68 \times 10^6$ | $3.38 \times 10^7$ | $4.43 \times 10^5$ | $5.90 \times 10^7$ |
| Shanxi | $3.14 \times 10^5$ | $4.14 \times 10^6$ | $4.42 \times 10^4$ | $2.26 \times 10^6$ |
| Inner Mongolia | $1.52 \times 10^4$ | $5.15 \times 10^5$ | $5.40 \times 10^3$ | $9.25 \times 10^4$ |
| Liaoning | $2.38 \times 10^5$ | $1.26 \times 10^6$ | $1.10 \times 10^4$ | $1.02 \times 10^6$ |
| jilin | $1.56 \times 10^5$ | $5.21 \times 10^5$ | $7.93 \times 10^3$ | $6.09 \times 10^5$ |
| Heilongjiang | $1.03 \times 10^5$ | $4.82 \times 10^5$ | $6.75 \times 10^3$ | $6.18 \times 10^5$ |
| Shanghai | $2.41 \times 10^3$ | $8.33 \times 10^4$ | $2.80 \times 10^3$ | $6.08 \times 10^5$ |
| Jiangsu | $6.67 \times 10^6$ | $1.72 \times 10^7$ | $3.14 \times 10^5$ | $6.88 \times 10^7$ |
| Zhejiang | $5.11 \times 10^3$ | $2.49 \times 10^4$ | $2.52 \times 10^2$ | $2.36 \times 10^5$ |
| Anhui | $1.16 \times 10^6$ | $1.24 \times 10^7$ | $9.39 \times 10^4$ | $1.12 \times 10^7$ |
| Fujian | $7.67 \times 10^0$ | $4.70 \times 10^1$ | $7.74 \times 10^{-1}$ | $1.77 \times 10^2$ |
| Jiangxi | $4.71 \times 10^5$ | $1.99 \times 10^6$ | $1.45 \times 10^4$ | $1.57 \times 10^6$ |
| Shandong | $2.82 \times 10^5$ | $3.88 \times 10^6$ | $5.05 \times 10^4$ | $6.02 \times 10^6$ |
| Henan | $6.39 \times 10^6$ | $2.67 \times 10^7$ | $3.93 \times 10^5$ | $4.28 \times 10^7$ |
| Hubei | $3.70 \times 10^5$ | $1.94 \times 10^6$ | $2.95 \times 10^4$ | $3.04 \times 10^6$ |
| Hunan | $1.75 \times 10^6$ | $7.00 \times 10^6$ | $5.90 \times 10^4$ | $5.84 \times 10^6$ |
| Guangdong | 0 | 0 | 0 | 0 |
| Guangxi | $8.05 \times 10^2$ | $4.37 \times 10^3$ | $2.67 \times 10^1$ | $2.34 \times 10^3$ |
| Hainan | 0 | 0 | 0 | 0 |
| Taiwan* | 0 | 0 | 0 | 0 |
| Chongqing | $6.46 \times 10^5$ | $2.05 \times 10^6$ | $1.94 \times 10^4$ | $1.50 \times 10^6$ |
| Sichuan | $4.55 \times 10^6$ | $1.59 \times 10^7$ | $1.37 \times 10^5$ | $2.16 \times 10^7$ |
| Guizhou | 0 | 0 | 0 | 0 |
| Yunnan | $1.13 \times 10^3$ | $6.78 \times 10^3$ | $6.16 \times 10^1$ | $3.69 \times 10^3$ |
| Tibet** | 0 | 0 | 0 | 0 |
| Shaanxi | $1.18 \times 10^5$ | $4.62 \times 10^5$ | $4.49 \times 10^3$ | $3.65 \times 10^5$ |
| Gansu | $6.71 \times 10^3$ | $1.83 \times 10^4$ | $1.22 \times 10^2$ | $4.12 \times 10^3$ |
| Qinghai | $5.56 \times 10^3$ | $1.82 \times 10^4$ | $1.96 \times 10^2$ | $3.96 \times 10^3$ |
| Ningxia | 0 | 0 | 0 | 0 |
| Xinjiang | $1.98 \times 10^4$ | $1.61 \times 10^5$ | $2.39 \times 10^3$ | $4.60 \times 10^4$ |

*,**, no data.