# Peer review of "A New Method for Calculating Highway Blocking due to High Impact Weather Conditions"

_Natural Hazards and Earth System Sciences, 2023_

## Referee Comment (RC1)

Review of

**A New Method for Calculating Highway Blocking due to High Impact Weather Conditions**

Author(s): Duanyang Liu, Tian Jing, Mingyue Yan, Ismail Gultepe, Yunxuan Bao, Hongbin Wang, Fan Zu

Manuscript No.: nhess-2023-230

Manuscript type: Research article

Special issue: Indirect and intangible impacts of natural hazards

**General comments**

This manuscript evaluated the characteristics of the highway-blocking event data in terms of meteorological conditions and spatiotemporal distribution. A 5-level classification of highway blocking was proposed. Finally, the authors developed a highway load index as a weighted average of a set of selected parameters to represent the loss due to highway blocking. The general methodology was promising but certain clarification is needed. The discussion was detailed while some rationale of critical decisions needs to be further elaborated. The conclusions were valid and provided insights for transport management authorities. Some editing for the English language is required throughout the manuscript.

The manuscript presents a study on a topic within the scope of the Natural Hazards and Earth System Sciences (NHESS). I would recommend this manuscript for publication after addressing the following comments with critical discussions and clarifications.

**Specific comments**

1. The highway-blocking features in Figure 2
   - The inner circle was not explained in the manuscript. Was there any causal relationship between the features of inner and outer circles?
   - Four main weather factors (i.e., fog, rainfall, snow, ice) were used in Table 1. However, ice was not shown in Figure 2, although it is mentioned in Line 170: "The highway blocking caused by snowfall (snow cover) and icing also accounts for 17% and 2%, respectively."
   - It is recommended to improve the pie chart by enlarging the circle and adding the indication line.
   - The discussion was around the four main weather factors. Please confirm that event caused by other weather factors were removed from the database.

2. The cluster analysis
   - It is recommended to elaborate on how the method was used in this study after the mathematical equations. For example, it is inferred from section 3.3 that the number of clusters were decided to be five, corresponding to the five levels? It is unclear what the input vectors were, were they 2-dimensional (i.e., blocking mileage and blocking time) or only the meteorological factors?

- What are the uncertainties of the clustering centers in Table 2? What is the sensitivity of the clustering centers with respect to the meteorological factors?
- Related to the previous point, as shown in Figure 6, the distribution of the 5-level highway blocking is dominated by Level 1 and Level 3, while Levels 4 are limited. Are these caused by the thresholds?
- Notation is unclear for $x_{i'}$ and $C(i')=k$
- Equation 2 "armin" should be "argmin"

3. The highway load index
   - The construction of highway load index was a weighted average of a set of selected indicators. This approach was similar to the widely used index construction using the principle component analysis (PCA). PCA takes advantage of multicollinearity and combines the highly correlated variables into a set of uncorrelated variables. In this study, the selected parameters were correlated, such as $\Delta TF_{load}$ the load capacity of freight transport for per kilometer and $TF_{load}$ the total freight transport, $\Delta P_{load}$ the number of people for per kilometer and $P_{load}$ the number of people. Please elaborate on why those parameters are selected and how the multicollinearity issue is addressed.
   - What were the impacts of highway blocking levels and high-impact weather conditions on the highway load index? If possible, please discussion on the disaggregation of the highway load index to highway blocking levels and high impact weather conditions.
   - Were the results aggregated across all the weather conditions? Some results for fog conditions are shown in Figure 8. What were the patterns for the other three weather conditions?
   - In Figure 8, given fog conditions, Jiangsu, Hebei, Henan and Sichuan show high levels of damage in terms of four economic indicators. Since most of the highway blocks were due to fog weather, it is expected in Figure 7 that these provinces would have a high level of highway load. However, the highest highway load occurred in Guangdong and Jiangsu only. Line 308-309: "No highway-blocking events caused by dense fog occur in Guangdong Province". What is the driving indicator behind the high level of highway load in Guangdong?

4. It is understood from Line 68-69: "Taiwan, Hong Kong and Macao are not included in this study due to the lack of data", which is also reflected in Table 1, 5 and 7 and Figure 3. However, for Figures 7 and 8, there were results for Tibet and Taiwan (coloured map). Please clarify how the results for Tibet and Taiwan are computed given that no data was available.

5. The CRITIC weigh method
   - The methodology was introduced in Section 2.3.2 and equation3-4 explained the computation of some key parameters. It is unclear which correlation coefficient $r^{ij}$ is used, Pearson correlation coefficient?
   - The result in Table 6 showed extra parameters however they are not discussed or explained in the main text. What was the importance of these parameters? What were the implications on the weights?
   - Line 131-135: "Contrast intensity, expressed as a standard deviation, indicates the dispersion degree of an indicator. The larger the standard deviation is, the greater the dispersion degree is, the larger the differences between samples are, and the larger the

assigned corresponding weights are." Suggested large standard deviation S leads to larger weights. However, this trend was not reflected in Table 6.

6. What is the time span of the data introduced in Section 2.2 Data source? What is the time resolution of the data, which is related 10-day window and hourly distribution in Figure 5?

7. Terminology and notations are not consistent.
   - Level 4 is referred to as the severe level between Line 245 and Line 265, where it is labelled as serious in Table 2
   - Please add notations $TF_{load}$, $GDP_{trans}$, $P_{load}$ and EP in the caption of Figure 8
   - "ice" vs "icing"
   - Equation 6 "GDPtrans the added value of the GDP generated by transportation" vs Table 5 "$GDPtrans$" unit "100 million yuan/km" which is normalised by length vs Table 7 "$GDPtrans$" unit "Ten thousand"

8. The English language needs to be improved. Some editing for the English language is required throughout the manuscript due to too many mistranslations or mistakes. The authors must seek the help of a native English-speaking person.
   - Line 55-56: "The ability to estimate highway traffic demand caused by the highway blocking during adverse weather events; therefore, it is critically needed." needs to be rephrased.
   - Table 2 five levels are recommended to be labelled as "slight-mild-moderate-severe-extreme".
   - Line 188-190: "There are large seasonal differences in highway blocking in various regions of China due to differences in geographical environment and climatic characteristics (Fig. 4a), and high-impact weather types (Fig. 4b), such as dense fog, snowfall (snow cover), rainfall (road slippery) and icing, **are also various**."
   - Line 232-235: "This study only considers the evaluation of road traffic by the blockage itself, and does not consider the basic resources of the road network and the impact of secondary disasters. If the road network resources are large, then the blocking may have little impact on the local road network, which is not considered in the blocking degree." What do the road network resources refer to?

9. Clarification is generally needed to better explain the motivation of critical decisions in this study.
   - Line 23-24: "Results suggested that the highway losses caused by dense fog was the main contributor for highway blocking conditions and occur at about 43%." 43% of what, the loss or the occurrence rate?
   - Line 81-82: "…follow the criteria of the Highway Traffic Blocking Information Submitting System of the Ministry of Transport of the People's Republic of China (2018, No. 451)." What are the criteria?
   - Line 89-90: "Therefore, all data were corrected in advance for spatio-temporal sequences, and the quality control was then carried out according to blocking causes and site descriptions, etc. 95% of the valid data is filtered out". Please elaborate on what are the correction procedure and quality control as large amounts of data are filtered out.

- Line 165: "$\alpha$, $\beta$, $\gamma$, $\delta$, $\varepsilon$, $\epsilon$, $\theta$, $\vartheta$ and $\mu$ are the corresponding coefficient values of each parameter." It is recommended to clarify that they were weights computed from the CRITIC method and referred to Table 6 as well.
- Line 231-232: "we select the blocking mileage (the distance of the highways blocking), blocking time and response time as the most crucial reference indicators." Why choose these three parameters? It is not clear where the response time is used in this study. What are the sensitivities of the results if more or less indicators were used?
- Line 236-238: "Firstly, the blocking mileage is used as the initial judgment condition of severity. Then, the blocking events caused by different meteorological factors are clustered. Finally, the severity of the blocking events is determined according to the size of the clustering centers." It is very unclear what is the procedure here and how the clustering was actually carried out.
- It is stated in Lines 280-290 that the normalised economic indicators were used in Equation 8. Thus, the notation in Equation 6 and 8 should be unified. For example, $H_d$ is the highway density, but it should be normalised highway density. Please confirm.
- Table 3 and 4, what does "/" indicate? No data or no obvious factors?

**Minor comments**

10. Please provide the link or reference to the data sources in Section 2.2.
11. Table 2 "type" is not sorted from A to E. Is there any specific reason or is it the cluster type that is not associated with the level?
12. Provinces in China are mentioned throughout the manuscript to discuss the highway blocking distribution due to high-impact weather conditions. Therefore, a map of provinces with labels is needed to facilitate the discussion.
13. Please add the subplot labels (e.g., (a) and (b)) to the figure, or use *Left* and *Right* in the caption
14. Line 54 please introduce HIW acronym before using it in the text. The first time you use the term, put the acronym in parentheses after the full term.
15. Figure 5 Left subplot a: results for Anhui was missing while results for Anhui was available in subplot b

---

## Referee Comment (RC2)

**Review of "A New Method for Calculating Highway Blocking due to High Impact Weather Conditions", by Liu et al. 2024 NHESSD**

**General comment**

In this study, the authors propose a new, data-driven approach to evaluating the impact of severe weather conditions on highway infrastructures across China. They suggest employing k-means and CRITIC methodologies to assess the significance of various natural phenomena and to classify the impact of highway interruptions using a set of significant variables (both technical and economic) provided by the authorities. The scope of the study is relevant and aligns with the journal's topics. However, there are critical aspects that prevent the publication of this version. Most notably, the methodological presentation requires significant improvement, necessitating more detailed explanations and changes to the current structure. Additionally, the presentation of results needs enhancement. Furthermore, the discussion and the adopted terminology require critical review. Finally, the English language usage needs improvement (a few examples are listed below, but an overall revision is recommended).

**Specific comments**

- Weather event definition:  the term "High-impact weather condition" lacks a specific and clear definition. It is not explicitly defined within the text, and its usage could be questionable. A natural event (i.e., rainfall, ice, etc.) might be severe or not, and might cause, or not, impacts on the highway and on other infrastructures depending on their characteristics. Therefore, I suggest referring simply to weather-related conditions. Furthermore, the adoption of this term implies that you consider exclusively such events, potentially overlooking medium-impact or low-impact weather conditions (which are not defined in the text). Perhaps, on page 2, line 50, you refer to "adverse" weather, which could serve as a suitable alternative.

- Data characterization:

  - Section 2.2.1: how are highway blockage events defined? What are the requirements in terms of their extent (length) or duration in order to be classified as such? Which is temporal and spatial aggregation of such events? Are those associated to a give region, district, or specific highway? Additional details should also be provided on the validity check you performed. The reference to "manual statistics" is not clear.
  - Section 2.2.2: how did you use the meteorological data to classifiy and attribute weather events to highway blocking scenarios? Which are the spatial and temporal aggregation you adopted for such data? How did you consider cases with multiple weather events attributions?
  - Section 2.2.3: More details need to be provided to describe the data characterizing the economic aspects of the highways. What temporal and spatial resolution do the data refer to? (e.g., annual flow, daily flow?) The sentence at L103 is not clear; what is its purpose and how are these classifications used?

- Abstract: there are several repetitions that could be removed, making the text more effective and clearer. Also, you should try to be quantitative while presenting the outcomes.

- Methodology:

- Section 2.3.1: although not new, the K-means methodology needs to be better explained. The analytical formalism is unclear. For instance, what are xi1 and xi2 of P4? I suggest rephrasing and expanding the overall section.
- Section 2.3.2: similarly, mathematical formalism should be checked (see perhaps eq. 3 where the same counter (i) is used for two summations having two reference set).
- Section 2.3.3: please define "express capacity". I believe a table summarizing the overall set of variables you adopted would benefit the methodology presentation. Which is the specific unit area that you adopted to calculate the highway density? L154-155: this sentence is definitely not clear: which are the items you are referring to? If there is strong correlation among some data, you should not consider all of them. Have you tried to perform a principal component analysis to evaluate such aspects?
  Also, lines 155-157 should go to previous section (2.3.1).

- Results

- The inner pie graph of figure 2 is not described within the text. Differences among the two plots are not clear, I suggest just keeping the external plot. Also, icing is not shown in the figure.
- Figure 3 is too small. I would consider to separate the two panels. Region names should be added to the map. Plots, as well as legends, should be more quantitative. Which are the values associated to the bars in pale 3b? Also, I suggest removing the lateral minor-boxes showing the islands, since it looks like there are nor results for those regions.
- L188-191 need to be rephrased.
- what do you mean for "basic resources of the road"? P10L233.
- Presentation of the severity classes should be moved to the methodological section. Specifically, text from the beginning of section 3.3 to line 247 is more appropriate in section 2. Then, in section 3 you present the results. Concerning such results, I suppose that the outcomes in Table 3 would be more readable if shown through graphs.
- Figure 7 is qualitative. I do not see much values on such plots unless, at least, upper and lower bounds are shown.
- Table 6: labels adopted for the items are different from those used within the text. Please revise and be uniform.

- Discussion

- The overall discussion is lacking, with insufficient critical analysis of the strengths and limitations of the methodology.
- The weights shown in eq. 8 are quite uniform among the considered items. What is the sensitivity of the outcomes to variations in such weights? Any comments on their values? It would have been beneficial to consider additional data or to remove some of those already considered.
- I disagree with the use of losses or damages, as adopted in the discussion and in Figure 8. From what I have understood, this analysis is an exposure evaluation rather than a damage assessment. The classifications shown in Figure 7 (which can be further explained with the details of Figure 8) provide a picture of the overall highway loads over the area, which do not necessarily correspond to real damage. This is a key aspect that deserves consideration and affects the overall scope and ambition of the manuscript. Furthermore, Conclusions should be revised accordingly to such definition.

**Minor comments:**

- P2, L(line)40-45: please check the sentence structures here and rephrase.
- P2, L54: HIW acronym is not defined.
- P4, L8: please check the sentence starting with "For the classification…". It looks uncomplete.

P2, L(line)40-45: please check the sentence structures here and rephrase.
P2, L54: HIW acronym is not defined.
P4, L8: please check the sentence starting with "For the classification…". It looks uncomplete.

---

## Referee Comment (RC3)

**Review of A New Method for Calculating Highway Blocking due to High Impact Weather Conditions**

Duanyang Liu1 , Tian Jing1, 2, Mingyue Yan3 , Ismail Gultepe4 , Yunxuan Bao1,2,5, Hongbin Wang1 , Fan Zu1

**General comments**

In this study, the authors present a methodology to evaluate the effects of different weather-related events on the Chinese highway network. While the statistical and geographical description of weather events and its effect on road impact is very interesting and well presented, several concepts that seem fundamental to understand the results, such as highway load and losses, are not clear and can be misleading. Better care should be given to clearly explain these concepts and separate them from their usual denominations for traffic demands and direct economic impacts.

Overall, the value proposition (why this study and its results are important) is not clear. The study should be reevaluated to clearly provide a valuable discussion of its results, considering the strengths and limitations of the methodology. Large revisioning for the English language and grammar is required throughout the manuscript, which is beyond the scope of the present review.

**Specific comments**

**Abstract**

The text in the abstract needs revision, I recommend to reorganize it considering a "why, how and what" storyline. "Why" focuses on the problem, "How" on the methodology and how it can help the problem, "What" focuses on the work done and the results. Currently, the Why is presented in a trivial way, the How is barely present and the What is too focused on the methods and not enough on the results and what they mean (the reference of 43% is not clear).

**Introduction**

The use of "high-impact weather conditions" is not ideal, it makes reference to impact and it is not clearly defined. I recommend focusing on the event and not the impact, adopting something similar to "adverse weather conditions". Also, it is best to not say "**the** high-impact weather conditions", just "high-impact weather conditions".

| Line 37 | I recommend the following change: "Therefore, driving in foggy weather is a potentially dangerous activity **for users, which increases the potential for** road blocks (Yan et al. 2014)." |
|---|---|
| Line 54 | the term HIW is used before it is described in line 57. Also, the study is not improving the effects on road blocking, but helping to better characterize and predict the impact. Please change the text to reflect that. |
| Line 55 | Statistics of China's road length and ranking needs a reference |
| Line 56 | "The ability to estimate highway traffic demand caused by the highway blocking during adverse weather events; therefore, it is critically needed." needs to be rewritten to clarify. I recommend:
 "Therefore, there is a critical need to improve the ability to estimate highway traffic impact, caused by highway blocking during adverse weather events" |
| Line 57 | Replace "factors" with "components" or "contributors" |
| Line 58 | Replace "affected" with "caused" |

**Data and methods**

- Please include a reference on where the authors are obtaining the data about the geometry and characteristics of the highway network.
- In Section 2.3.3, it is unclear to me how the K-means algorithm is applied to the methodology proposed.
- Please provide some lines into what is the physical meaning of the CRITIC weights in this context. This will help clarify the percentages reported in the results section.

**Results**

- The bar plots in Figure 3.b need to be clarified to give a reference of what does the height of the bars represent.
- Given the type of events considered (fog, rain, snow and ice) it is not clear to me why so many instances of road blocking are happening in the summer. Please include some lines to acknowledge and clarify this.
- The way that the severity of the blocking is defined needs to be further clarified. It is not clear to me what "blocking mileage" (L) is. Maybe including a couple of illustrative examples would help.

**Discussions**

- The way that the concept of "highway load" is being treated in this section is not clear to me, or how it relates to losses. While the idea of using a "highway load" as a proxy for economic impact potential is clear, it is misleading to say they directly relate to economic losses. There are many other factors that come into play when evaluating economic losses. It is advised to clarify this concept, to assure the reader what the methodology is actually capturing and what the results mean potentially.
- The results presented in Figure 8 should be first presented in the Results section, clearly explaining how they were derived and only then can they be discussed in this section. In the current format it is not clear where these values are coming from.
- Overall, this section lacks transparency about the strengths and limitations of the methodology.

**Conclusions**

- The final parts of this section makes reference to natural disasters, which have not been clearly discussed previously. I suggest this is removed to avoid misleading the readers into thinking that these events were accounted for in this study.
- There is a mention to direct and indirect losses, though their difference has not been discussed or defined in the paper, I suggest to remove it.

---

## Author Comment (AC1)

Dear Editor and Reviewer,

Thank you very much for the constructive suggestions on our paper entitled "A New Method for Calculating Highway Blocking due to High Impact Weather Conditions". They are very helpful for improving the manuscript. We have revised the manuscript accordingly. We sincerely hope that you will find this version acceptable to be published in Natural Hazards and Earth System Sciences.

Best regards.

Yours sincerely,

Duanyang Liu (Corresponding Author)

Nanjing Joint Institute for Atmospheric Sciences (NJIAS)

E-mail: liuduanyang@cma.gov.cn;

**Manuscript #    NHESS-2023-230**

Manuscript Title: A New Method for Calculating Highway Blocking due to High Impact Weather Conditions

**Responses to Reviewer #1**

**General comments**

*This manuscript evaluated the characteristics of the highway-blocking event data in terms of meteorological conditions and spatiotemporal distribution. A 5-level classification of highway blocking was proposed. Finally, the authors developed a highway load index as a weighted average of a set of selected parameters to represent the loss due to highway blocking. The general methodology was promising but certain clarification is needed. The discussion was detailed while some rationale of critical decisions needs to be further elaborated. The conclusions were valid and provided insights for transport management authorities. Some editing for the English language is required throughout the manuscript.*

*The manuscript presents a study on a topic within the scope of the Natural Hazards and Earth System Sciences (NHESS). I would recommend this manuscript for publication after addressing the following comments with critical discussions and clarifications.*

**RESPONSE: Thank you for your suggestions.**
* * *
**Specific comments**

*1、 The highway-blocking features in Figure 2*

● *The inner circle was not explained in the manuscript. Was there any causal relationship between the features of inner and outer circles?*

● *Four main weather factors (i.e., fog, rainfall, snow, ice) were used in Table 1. However, ice was not shown in Figure 2, although it is mentioned in Line 170: "The highway blocking caused by snowfall (snow cover) and icing also accounts for 17% and 2%, respectively."*

● *It is recommended to improve the pie chart by enlarging the circle and adding the indication line.*

● *The discussion was around the four main weather factors. Please confirm that event caused by other weather factors were removed from the database.*

**RESPONSE: Revised.The inner pie graph of figure 2 is deleted.**
* * *
2、 The cluster analysis

● *It is recommended to elaborate on how the method was used in this study after the mathematical equations. For example, it is inferred from section 3.3 that the number of clusters were decided to be five, corresponding to the five levels? It is unclear what the input vectors were, were they 2-dimensional (i.e., blocking mileage and blocking time) or only the meteorological factors?*

**RESPONSE: Revised. Firstly, the blocking mileage is used as the initial judgment condition of severity. Then, using equatoin 2, the blocking events caused by different meteorological factors are clustered. the blocking mileage, blocking time and response time in different high-impact weathers asl the input vectors**

in the calculations. Finally, the severity of the blocking events is determined according to the size of the clustering centers.
* * *
● *What are the uncertainties of the clustering centers in Table 2? What is the sensitivity of the clustering centers with respect to the meteorological factors?*

**RESPONSE: Revised.**
**The "uncertain center" in the K-means algorithm mainly refers to the fact that at the beginning of the algorithm, the centers of the clusters (i.e., the centers of the clusters) are randomly selected, first the number of clusters K needs to be determined, and then K data points are randomly selected from the dataset to be the initial center of the clusters. The selection of these initial centroids is random and there are no fixed rules or criteria.**
**Such uncertain centers are selected by categorization only by calculating the blocking intensity expressed by blocking time \* blocking mileage, and all blocking events occur because of weather factors, so sensitivity analysis cannot be performed.**
* * *
● *Related to the previous point, as shown in Figure 6, the distribution of the 5-level highway blocking is dominated by Level 1 and Level 3, while Levels 4 are limited. Are these caused by the thresholds?*

**RESPONSE: Revised.**
**This was analyzed by a clustering algorithm, in which the clustering of blocking intensities showed a low distribution of high-level blocking events; usually disasters in which a specific long period of time and a wide range of meteorological hazards do not often occur in daily life.**
**This is a presentation of the clustering results, sudden and prolonged disasters are not common, so the high level of highway blockage is less, in most cases, small-scale unexpected weather affects the smoothness of the traffic, but the highway management will also be able to deal with the problem as soon as possible, when it comes to the natural disasters of high intensity and wide range of the highway management can't unblock the traffic as soon as possible, therefore the intensity of the blockage events in this case will be very high, so the high level of the clustering results will be highlighted and present a scanty number of results.**
* * *
● *Notation is unclear for xi' and C(i')=k*

**RESPONSE: Revised.**
* * *
● *Equation 2 "armin" should be "argmin"*

**RESPONSE: Revised.The inner pie graph of figure 2 is deleted.**
* * *
*3、 The highway load index*

● *The construction of highway load index was a weighted average of a set of selected indicators. This approach was similar to the widely used index construction using the principle component analysis (PCA). PCA takes advantage of multicollinearity and combines the highly correlated variables into a set of uncorrelated variables. In this study, the selected parameters were correlated, such as ΔTFload the load capacity of freight transport for per kilometer and TFload the total freight transport, Δ load the number of people for per kilometer and   load the number of people. Please elaborate on why those parameters are selected and how the multicollinearity issue is addressed.*

**RESPONSE: Revised. The CRITIC method is an objective evaluation method based on the comparative strength of the evaluation indicators, which fully utilizes the objective attributes of the data itself to carry out scientific evaluations. The CRITIC method takes into account the size of the variability of the indicators while also taking into account the correlation between the indicators. This means that it not only focuses on the volatility of individual**

indicators, but also takes into account the mutual influence and conflict between indicators, thus improving the comprehensiveness and accuracy of the evaluation. Therefore, we chose the composite and unit indicators for evaluation, which are also more suitable for this method.
* * *
- *What were the impacts of highway blocking levels and high-impact weather conditions on the highway load index? If possible, please discussion on the disaggregation of the highway load index to highway blocking levels and high impact weather conditions.*

RESPONSE: Revised. We carry out the assessment of the carrying pressure per unit length of road in different areas by calculating the highway loading index, and when it carries more transportation tasks, more losses will be incurred in the event of a blockage.
* * *
- *Were the results aggregated across all the weather conditions? Some results for fog conditions are shown in Figure 8. What were the patterns for the other three weather conditions?*

RESPONSE: Revised.

In the result part, we introduce the calculation process of the new calculation method in detail. In the discussion part, we choose the typical high impact weather for detailed discussion.

By carrying out this new calculation method proposed, we have analyzed and calculated all four types of catastrophic weather, and because the results are too lengthy, we have selected the fog weather blocking loss distribution, which has the highest percentage, for a demonstration. If needed, we can analyze the damage caused by the remaining three hazardous weather types of weather in a presentation and discussion.
* * *
- *In Figure 8, given fog conditions, Jiangsu, Hebei, Henan and Sichuan show high levels of damage in terms of four economic indicators. Since most of the highway blocks were due to fog weather, it is expected in Figure 7 that these provinces would have a high level of highway load. However, the highest highway load occurred in Guangdong and Jiangsu only. Line 308-309: "No highway-blocking events caused by dense fog occur in Guangdong Province". What is the driving indicator behind the high level of highway load in Guangdong?*

RESPONSE: Revised. Add this sentence: Guangdong Province, as an economically developed province in China, and near the sea has many harbors. Guangdong province highway and other infrastructure construction for a long time, basically covering the entire province, not only to bear the pressure of the port cargo to inland, but also to bear the pressure of the entire province manufacturing transportation. So the highway load in Guangdong are in the high level.
* * *
4、 *It is understood from Line 68-69: "Taiwan, Hong Kong and Macao are not included in this study due to the lack of data", which is also reflected in Table 1, 5 and 7 and Figure 3. However, for Figures 7 and 8, there were results for Tibet and Taiwan (coloured map). Please clarify how the results for Tibet and Taiwan are computed given that no data was available.*

RESPONSE: Figures 7 and 8 are Revised. In Figures 7 and 8, the results for Tibet and Taiwan (white, no data)
* * *
5、 *The CRITIC weigh method*
- *The methodology was introduced in Section 2.3.2 and equation3-4 explained the computation*

*of some key parameters. It is unclear which correlation coefficient rij is used, Pearson correlation coefficient?*

**RESPONSE: Equation3-4 and the key parameters are Revised.**

**Contrast intensity, expressed as a standard deviation, indicates the dispersion degree of an indicator. The larger the standard deviation is, the greater the dispersion degree is, the larger the differences between samples are, and the larger the assigned corresponding weights are. The standard deviation *S* can be expressed in Eq. (3).**

$$S_j = \sqrt{\frac{\sum_{i=1}^{p}\left(x_{ij}-\frac{1}{n}\sum_{i=1}^{n}x_{ij}\right)^2}{n-1}} \tag{3},$$

**where $x_{ij}$ denotes the data processed by standard deviation, $S_j$ the standard deviation of the *j*th indicator, *n* the total number of samples, and *p* the total number of indicators.**

**Correlation is expressed as the correlation coefficient between indicators. The stronger the correlation between indicators is, the higher the repetition rate of information expression. Therefore, the corresponding weights of the indicators can be reduced to a certain extent. The correlation coefficient *R* can be expressed in Eq. (4).**

$$R_j = \sum_{i=1}^{p}(1-r_{ij}) \tag{4},$$

**where, $R_j$ indicates the correlation coefficients of the *j*th indicator with the other indicators, and $r_{ij}$ denotes the correlation coefficient of the *i*th indicator with the *j*th indicator.**
* * *
- *The result in Table 6 showed extra parameters however they are not discussed or explained in the main text. What was the importance of these parameters? What were the implications on the weights?*

**RESPONSE: In the CRITIC weighting method, although the standard deviation, as a measure of the intensity of comparison, reflects the magnitude of the difference in the values of the same indicator between different evaluation objects, i.e., the volatility, the magnitude of the standard deviation is not the only factor that directly determines the magnitude of the weights. The determination of weights also needs to take into account the conflicting nature of different indicators, i.e., the correlation between them.**

**Specifically: when an indicator has a large standard deviation but also a high correlation with other indicators, its weight may be weakened by a high correlation (i.e., low conflictivity).**

**Conversely, when an indicator has a low correlation with other indicators (i.e., high conflictivity), although not the largest standard deviation, its weight may be relatively high because of the unique information it provides.**

**We therefore present both mutability and standard deviation in a table.**

- *Line 131-135: "Contrast intensity, expressed as a standard deviation, indicates the dispersion degree of an indicator. The larger the standard deviation is, the greater the dispersion degree is, the larger the differences between samples are, and the larger the corresponding weights are." Suggested large standard deviation S leads to larger weights. However, this trend was not reflected in Table 6.*

**RESPONSE:**

**In the CRITIC weighting method, although the standard deviation, as a measure of the intensity of comparison, reflects the magnitude of the difference in the values of the same indicator between different evaluation objects, i.e., the volatility, the magnitude of the standard deviation is not the only factor that directly determines the magnitude of the weights.**

**The determination of weights also needs to take into account the conflicting nature of different indicators, i.e., the correlation between them.**

**Specifically: when an indicator has a large standard deviation but also a high correlation with other indicators, its weight may be weakened by a high correlation (i.e., low conflictivity).**

**Conversely, when an indicator has a low correlation with other indicators (i.e., high conflictivity), although not the largest standard deviation, its weight may be relatively high because of the unique information it provides.**

6、 *What is the time span of the data introduced in Section 2.2 Data source? What is the time resolution of the data, which is related 10-day window and hourly distribution in Figure 5?*

**RESPONSE: Revised. Add sentences:**

**Highway blockage data comes from Highway Monitoring & Emergency Response Center, and the data recording process and specifications are in accordance with the "Information Reporting System for Highway Traffic Blockage of the Ministry of Transportation and Communications of the People's Republic of China" issued by the Ministry of Transportation and Communications of the People's Republic of China.The dataset contains 16 indicators: province name, submitting department, route name, route number, starting and ending pile number (Highway pile numbers are usually combined with the milestone system and are expressed in K kilometers ± meters. That is, along the direction of the road, the pile number at the starting point is k0+000, and one pile number is marked every certain distance (such as 100 meters), and the corresponding place is marked), reasons of highway blocking, blocking mileage (the distance of the highways blocking), status, blocking type, information event classification, site description, disposal measure, time of finding blockage, submitting time, expected recovery time, and actual recovery time. Since highway blocking information is submitted by manual statistics, there is a possibility of manual statistical errors. Therefore, all data were pre-corrected with a time series correction and then verified based on the cause of the blockage and the meteorological data of the station at the time, among other things. Quality control resulted in the retention of 95% of valid data.**
* * *
7、 *Terminology and notations are not consistent.*

● *Level 4 is referred to as the severe level between Line 245 and Line 265, where it is labelled as serious in Table 2*

**RESPONSE: Revised. Table 2 five levels are recommended to be labelled as "slight-mild-moderate-severe-extreme".**
* * *
● *Please add notations TFload, GDPtrans, Pload and EP in the caption of Figure 8*

**RESPONSE: Revised. Table 2 five levels are recommended to be labelled as "slight-mild-moderate-severe-extreme".**
* * *
● *"ice" vs "icing"*

**RESPONSE:Revised.**
* * *
● *Equation 6 "GDPtrans the added value of the GDP generated by transportation" vs Table 5 "   " unit "100 million yuan/km" which is normalised by length vs Table 7 "   " unit "Ten thousand"*

**RESPONSE: Table 5, table 7 Revised..**
* * *
8、 The English language needs to be improved. Some editing for the English language is required throughout the manuscript due to too many mistranslations or mistakes. The authors must seek the help of a native English-speaking person.

● *Line 55-56: "The ability to estimate highway traffic demand caused by the highway blocking*

*during adverse weather events; therefore, it is critically needed." needs to be rephrased.*

**Response:Revised. Therefore, there is a critical need to improve the ability to estimate highway traffic, caused by the highway blocking during adverse weather events.**
* * *
- *Table 2 five levels are recommended to be labelled as "slight-mild-moderate-severe-extreme".*

**Response:Revised.**
* * *
- *Line 188-190: "There are large seasonal differences in highway blocking in various regions of China due to differences in geographical environment and climatic characteristics (Fig. 4a), and high-impact weather types (Fig. 4b), such as dense fog, snowfall (snow cover), rainfall (road slippery) and icing, are also various."*

**Response:Revised. "There are large seasonal differences in highway blocking in various regions of China due to differences in geographical environment and climatic characteristics (Fig. 4a), and high-impact weather types (Fig. 4b), such as dense fog, snowfall (snow cover), rainfall (road slippery) and icing, etc."**

- *Line 232-235: "This study only considers the evaluation of road traffic by the blockage itself, and does not consider the basic resources of the road network and the impact of secondary disasters. If the road network resources are large, then the blocking may have little impact on the local road network, which is not considered in the blocking degree." What do the road network resources refer to?*

**Response:Revised. the road miles per unit area, the higher the density of road network per unit area, the more abundant road network resources.**
* * *
*9、Clarification is generally needed to better explain the motivation of critical decisions in this study.*

- *Line 23-24: "Results suggested that the highway losses caused by dense fog was the main contributor for highway blocking conditions and occur at about 43%." 43% of what, the loss or the occurrence rate?*

**RESPONSE: Revised. These lines are deleted.**
* * *
- *Line 81-82: "...follow the criteria of the Highway Traffic Blocking Information Submitting System of the Ministry of Transport of the People's Republic of China (2018, No. 451)." What are the criteria?*

**RESPONSE: Revised. The highway-blocking events data obtained from the Ministry of Transport of the People's Republic of China (Fig. 1a) follow the criteria of the Highway Traffic Blocking Information Submitting System of the Ministry of Transport of the People's Republic of China (2018, No. 451;https://www.hunan.gov.cn/xxgk/wjk/zcfgk/202007/t20200730_e1c6436a-6aff-43d0-9c74-c0822311b8db.html).**
* * *
- *Line 89-90: "Therefore, all data were corrected in advance for spatio-temporal sequences, and the quality control was then carried out according to blocking causes and site descriptions, etc. 95% of the valid data is filtered out". Please elaborate on what are the correction procedure and quality control as large amounts of data are filtered out.*

**RESPONSE: Revised. Therefore, all data were pre-corrected with a time series correction and then verified based on the cause of the blockage and the meteorological data of the station at the time, among other things. Quality control resulted in the retention of 95% of valid data.**
* * *
- *Line 165: "α, β, γ, δ, ε, ϵ, θ, ϑ and μ are the corresponding coefficient values of each parameter." It is recommended to clarify that they were weights computed from the CRITIC method and referred to Table 6 as well.*

**RESPONSE: Revised. α, β, γ, δ, ε, ϵ, θ, ϑ and μ are the corresponding coefficient values of each parameter, these parameters will be computed according the above data, and these will be detailed calculated in the results part.**

- *Line 231-232: "we select the blocking mileage (the distance of the highways blocking), blocking time and response time as the most crucial reference indicators." Why choose these three parameters? It is not clear where the response time is used in this study. What are the sensitivities of the results if more or less indicators were used?*

**RESPONSE: Revised. blocking mileage (the distance of the highways blocking), blocking time and response time are three characteristic quantities that reflect highway blocking, so we use three indicators for analysis.**
* * *
- *Line 236-238: "Firstly, the blocking mileage is used as the initial judgment condition of severity. Then, the blocking events caused by different meteorological factors are clustered. Finally, the severity of the blocking events is determined according to the size of the clustering centers." It is very unclear what is the procedure here and how the clustering was actually carried out.*

**RESPONSE: Revised. Firstly, the blocking mileage is used as the initial judgment condition of severity. Then, using equatoin 2, the blocking events caused by different meteorological factors are clustered. the blocking mileage, blocking time and response time in different high-impact weathers asl the input vectors in the calculations. Finally, the severity of the blocking events is determined according to the size of the clustering centers.**
* * *
- *It is stated in Lines 280-290 that the normalised economic indicators were used in Equation 8. Thus, the notation in Equation 6 and 8 should be unified. For example, Hd is the highway density, but it should be normalised highway density. Please confirm.*

**Response: All the economic indicators were normalised before used.**
* * *
- *Table 3 and 4, what does "/" indicate? No data or no obvious factors?*

**Response: Revised.   "/" assumes that there are no blocking events of this type at this level.**
**The data were standardised at the pre-processing stage, where Equation 6, which was used in the methodology presentation, is a methodological scenario and is therefore represented using Arabic letters. Equation 8 is a presentation of the final calculations.**
* * *
**Minor comments**

*10、      Please provide the link or reference to the data sources in Section 2.2.*

**Response: Revised. The highway-blocking events data obtained from the Ministry of Transport of the People's Republic of China (Fig. 1a) follow the criteria of the Highway Traffic Blocking Information Submitting System of the Ministry of Transport of the People's Republic of China (2018, No. 451; https://www.hunan.gov.cn/xxgk/wjk/zcfgk/202007/t20200730_e1c6436a-6aff-43d0-9c74-c0822311b8db.html) .**
* * *
*11、      Table 2 "type" is not sorted from A to E. Is there any specific reason or is it the cluster type that is not associated with the level?*

**Response: The type from A to E are deleted.**
* * *
*12、 Provinces in China are mentioned throughout the manuscript to discuss the highway blocking distribution due to high-impact weather conditions. Therefore, a map of provinces with labels is needed to facilitate the discussion.*

**Response: Revised.**
* * *
*13、 Please add the subplot labels (e.g., (a) and (b)) to the figure, or use Left and Right in the caption*

**Response: Revised.**
* * *
*14、 Line 54 please introduce HIW acronym before using it in the text. The first time you use the term, put the acronym in parentheses after the full term.*

Response: Revised. The HIW acronym introduced in Abstract.
* * *
*15、 Figure 5 Left subplot a: results for Anhui was missing while results for Anhui was available in subplot b*

**Response: Revised.**

---

## Author Comment (AC2)

Dear Editor and Reviewer,

Thank you very much for the constructive suggestions on our paper entitled "A New Method for Calculating Highway Blocking due to High Impact Weather Conditions". They are very helpful for improving the manuscript. We have revised the manuscript accordingly. We sincerely hope that you will find this version acceptable to be published in Natural Hazards and Earth System Sciences.

Best regards.

Yours sincerely,

Duanyang Liu (Corresponding Author)

Nanjing Joint Institute for Atmospheric Sciences (NJIAS)

E-mail: liuduanyang@cma.gov.cn;

**Manuscript #   NHESS-2023-230**

Manuscript Title: A New Method for Calculating Highway Blocking due to High Impact Weather Conditions

**Responses to Reviewer #2**

**General comment**

*In this study, the authors propose a new, data-driven approach to evaluating the impact of severe weather conditions on highway infrastructures across China. They suggest employing k-means and CRITIC methodologies to assess the significance of various natural phenomena and to classify the impact of highway interruptions using a set of significant variables (both technical and economic) provided by the authorities. The scope of the study is relevant and aligns with the journal's topics. However, there are critical aspects that prevent the publication of this version. Most notably, the methodological presentation requires significant improvement, necessitating more detailed explanations and changes to the current structure. Additionally, the presentation of results needs enhancement. Furthermore, the discussion and the adopted terminology require critical review. Finally, the English language usage needs improvement (a few examples are listed below, but an overall revision is recommended).*

**RESPONSE: Thank you for your suggestions.**
* * *
**Specific comments**

*- Weather event definition:*

*the term "High-impact weather condition" lacks a specific and clear definition. It is not explicitly defined within the text, and its usage could be questionable. A natural event (i.e., rainfall, ice, etc.) might be severe or not, and might cause, or not, impacts on the highway and on other infrastructures depending on their characteristics. Therefore, I suggest referring simply to weather-related conditions. Furthermore, the adoption of this term implies that you consider exclusively such events, potentially overlooking medium-impact or low-impact weather conditions (which are not defined in the text). Perhaps, on page 2, line 50, you refer to "adverse" weather, which could serve as a suitable alternative.*

**RESPONSE: Thank you for your suggestions. The World Meteorological Organization (WMO) defines high-impact weather as severe weather events that have significant adverse impacts on society, infrastructure, and the environment. These events can cause widespread damage, disruption, and loss of life (Marsigli, 2021).**

**High-impact weather refers to severe weather events that have a significant impact on human activities, infrastructure, and the environment. These events can cause widespread damage, disruption, and loss of life. In this study, we discuss the high impact weather that has an impact on highway operation.**
* * *
*- Data characterization:*

- *Section 2.2.1: how are highway blockage events defined? What are the requirements in terms of their extent (length) or duration in order to be classified as such? Which is temporal and spatial aggregation of such events? Are those associated to a give region, district, or specific*

*highway? Additional details should also be provided on the validity check you performed. The reference to "manual statistics" is not clear.*

**RESPONSE: Revised. Add sentences:**

**A highway-blocking events is a state in which a highway is impassable or forced to close for some reason. Depending on the nature and duration of the blockage, it can be divided into two categories: planned and unexpected. Planned blockages include those caused by planned events such as highway maintenance and construction, reconstruction and expansion, and major social activities. Sudden-type blockages include sudden highway blockages caused by natural disasters (such as geological disasters, severe weather, etc.), accidents and disasters, public health incidents, social security incidents and other reasons. In this study, highway blockage under the influence of natural disasters and weather is selected as the main body of the study.**

**Highway blockage data comes from Highway Monitoring & Emergency Response Center, and the data recording process and specifications are in accordance with the "Information Reporting System for Highway Traffic Blockage of the Ministry of Transportation and Communications of the People's Republic of China" issued by the Ministry of Transportation and Communications of the People's Republic of China. Therefore, all data were pre-corrected with a time series correction and then verified based on the cause of the blockage and the meteorological data of the station at the time, among other things. Quality control resulted in the retention of 95% of valid data.**
* * *
- *Section 2.2.2: how did you use the meteorological data to classify and attribute weather events to highway blocking scenarios? Which are the spatial and temporal aggregation you adopted for such data? How did you consider cases with multiple weather events attributions?*

**RESPONSE: Revised. Add sentences:**

**According to the time and place of the expressway blockage, the meteorological observation data of this area during this period are checked with the weather events recorded by the observers to ensure the consistency of the data. In the case of multiple weather phenomena, we refer to the one recorded by highway blocking recaroded data.**
* * *
- *Section 2.2.3: More details need to be provided to describe the data characterizing the economic aspects of the highways. What temporal and spatial resolution do the data refer to? (e.g., annual flow, daily flow?) The sentence at L103 is not clear; what is its purpose and how are these classifications used?*

**RESPONSE: Revised. Add sentences: The above data are the traffic per 10 kilometers of the highway, per month(The industry classification of transportation is based on the industry classification of national economic activities, https://data.stats.gov.cn/easyquery.htm?cn=C01). The above data will be calculated and applied in 2.3.3 Calculation method of highway load.**
* * *
*- Abstract: there are several repetitions that could be removed, making the text more effective and clearer. Also, you should try to be quantitative while presenting the outcomes.*

**RESPONSE: Revised.**
* * *
*- Methodology:*
- *Section 2.3.1: although not new, the K-means methodology needs to be better explained. The analytical formalism is unclear. For instance, what are xi1 and xi2 of P4? I suggest rephrasing and expanding the overall section.*

**RESPONSE: Revised.**

**K-means clustering is an unsupervised machine learning method without prior knowledge (that is, no classification criteria is given before classification). The goal of this algorithm is to find groups in the data, with the number of groups represented by the variable K. One chooses the desired number of clusters, and the K-means procedure iteratively moves the centers to minimize the total within-cluster variance. Specifically, the criterion is minimized by assigning the observations to the K clusters in such a way that within each cluster the average dissimilarity of the observations from the cluster mean, as defined by the points in that cluster, is minimized (Hastie et al., 2009).**
* * *
- Section 2.3.2: similarly, mathematical formalism should be checked (see perhaps eq. 3 where the same counter (i) is used for two summations having two reference set).

**RESPONSE: Equation3-4 and the key parameters are Revised.**

**Contrast intensity, expressed as a standard deviation, indicates the dispersion degree of an indicator. The larger the standard deviation is, the greater the dispersion degree is, the larger the differences between samples are, and the larger the assigned corresponding weights are. The standard deviation $S$ can be expressed in Eq. (3).**

$$S_j = \sqrt{\frac{\sum_{i=1}^{p}\left(x_{ij}-\frac{1}{n}\sum_{i=1}^{n}x_{ij}\right)^2}{n-1}} \tag{3},$$

**where $x_{ij}$ denotes the data processed by standard deviation, $S_j$ the standard deviation of the $j$th indicator, $n$ the total number of samples, and $p$ the total number of indicators.**

**Correlation is expressed as the correlation coefficient between indicators. The stronger the correlation between indicators is, the higher the repetition rate of information expression. Therefore, the corresponding weights of the indicators can be reduced to a certain extent. The correlation coefficient $R$ can be expressed in Eq. (4).**

$$R_j = \sum_{i=1}^{p}(1-r_{ij}) \tag{4},$$

**where, $R_j$ indicates the correlation coefficients of the $j$th indicator with the other indicators, and $r_{ij}$ denotes the correlation coefficient of the $i$th indicator with the $j$th indicator.**
* * *
- Section 2.3.3: please define "express capacity". I believe a table summarizing the overall set of variables you adopted would benefit the methodology presentation. Which is the specific unit area that you adopted to calculate the highway density? L154-155: this sentence is definitely not clear: which are the items you are referring to? If there is strong correlation among some data, you should not consider all of them. Have you tried to perform a principal component analysis to evaluate such aspects?

**RESPONSE: Revised.**
**The adopted highway density is the sum of highway mileage per unit square kilometer, and "express capacity" is the volume of express delivery, which includes not only parcels, but also documents and other types of documents, and also plays an important role in information transmission, so we chose it as one of the reference quantities.**
**The use of highly correlated data is to better fit the CRITIC method.The CRITIC method combines two dimensions, intensity of comparison and conflict, to combine the weights of indicators. Comparative strength is expressed using the standard deviation, which reflects the degree of dispersion of data within the indicator, and conflict is expressed using the correlation coefficient, which reflects the correlation between indicators.**
* * *
- Also, lines 155-157 should go to previous section (2.3.1).

**RESPONSE: they are not the same methods. section (2.3.1) is clustering analysis, and 2.3.3 is CRITIC .**
* * *
- Results
- The inner pie graph of figure 2 is not described within the text. Differences among the two plots are not clear, I suggest just keeping the external plot. Also, icing is not shown in the figure.

**RESPONSE: Revised. The inner pie graph of figure 2 is deleted.**
* * *
- • Figure 3 is too small. I would consider to separate the two panels. Region names should be added to the map. Plots, as well as legends, should be more quantitative. Which are the values

*associated to the bars in pale 3b? Also, I suggest removing the lateral minor-boxes showing the islands, since it looks like there are nor results for those regions.*
**RESPONSE: Revised. The inner pie graph of figure 2 is deleted.**
* * *
- *L188-191 need to be rephrased.*

**Response:Revise. "There are large seasonal differences in highway blocking in various regions of China due to differences in geographical environment and climatic characteristics (Fig. 4a), and high-impact weather types (Fig. 4b), such as dense fog, snowfall (snow cover), rainfall (road slippery) and icing, etc."**
* * *
- *what do you mean for "basic resources of the road"? P10L233.*

**Response:Revised. the road miles per unit area, the higher the density of road network per unit area, the more abundant road network resources.**
* * *
- *Presentation of the severity classes should be moved to the methodological section. Specifically, text from the beginning of section 3.3 to line 247 is more appropriate in section 2. Then, in section 3 you present the results. Concerning such results, I suppose that the outcomes in Table 3 would be more readable if shown through graphs.*

**Response:We tried to carry out Tables 3 and 4 to present them with graphs, but found that, with more information, the results were not satisfactory, and the tables can be carried out to quantify the characteristics of different regions in different seasons.**
* * *
- *Figure 7 is qualitative. I do not see much values on such plots unless, at least, upper and lower bounds are shown.*

**Response:Revised.**
* * *
- *Table 6: labels adopted for the items are different from those used within the text. Please revise and be uniform.*

**Response:Revised.**
* * *
*- Discussion*
- *The overall discussion is lacking, with insufficient critical analysis of the strengths and limitations of the methodology.*

**Response:Revised.**
**In order to assess the economic losses caused by highway blockages due to high-impact weather events, we previously collected data and established a model for evaluating economic losses. We then used this model to calculate the economic losses caused by highway blockages resulting from high-impact weather events. Below, we compare specific high-impact weather events with actual loss data to verify the accuracy and reliability of our assessment model.**
* * *
- The weights shown in eq. 8 are quite uniform among the considered items. What is the sensitivity of the outcomes to variations in such weights? Any comments on their values? It would have been beneficial to consider additional data or to remove some of those already considered.

**Response:in eq. 8, Hd the highway density, ΔTFload the load capacity of freight transport for per kilometer, GDPtrans the added value of the GDP generated by transportation, ΔPload the number of people for per kilometer, VD the vehicle density, EP the number of express packages, TFload the total freight transport,**

**Pload the number of people, and Vprivate the number of private vehicles. These parameters are all related to the economic losses**
* * *
- *I disagree with the use of losses or damages, as adopted in the discussion and in Figure 8. From what I have understood, this analysis is an exposure evaluation rather than a damage assessment. The classifications shown in Figure 7 (which can be further explained with the details of Figure 8) provide a picture of the overall highway loads over the area, which do not necessarily correspond to real damage. This is a key aspect that deserves consideration and affects the overall scope and ambition of the manuscript. Furthermore, Conclusions should be revised accordingly to such definition.*

Response:Thank you for you suggestions. The economic parameters are calculated in eq.8, thus the results are using the exposure evaluation to estimate the economic loss parameters.
* * *
**Minor comments:**
- *P2, L(line)40-45: please check the sentence structures here and rephrase.*

Response: Revised. The road surface with snow or icing can lead to slower vehicle speeds and a decrease in fuel combustion efficiency (Hallegatte 2008; Min et al., 2016). The work of Min et al. (2016) showed that when 10% improvement occurs in road surface conditions, 0.6–2% reduction in air emissions amount can occur. The weather forecasting products, if used, the road transportation sectors can generate great economic benefit. Frei et al. (2014) found that the use of meteorology in the road transportation sector in Switzerland generates an economic benefit to the national economy 75.1-91.2 million U.S. dollars (cost/benefit ratio of around as 1:10).
* * *
- *P2, L54: HIW acronym is not defined.*

Response:Revised.
* * *
- *P4, L8: please check the sentence starting with "For the classification…". It looks uncomplete.*

Response:Revised. The goal of this algorithm is to find groups in the data, with the number of groups represented by the variable K.

---

## Author Comment (AC3)

Dear Editor and Reviewer,

Thank you very much for the constructive suggestions on our paper entitled "A New Method for Calculating Highway Blocking due to High Impact Weather Conditions". They are very helpful for improving the manuscript. We have revised the manuscript accordingly. We sincerely hope that you will find this version acceptable to be published in Natural Hazards and Earth System Sciences.

Best regards.

Yours sincerely,

Duanyang Liu (Corresponding Author)

Nanjing Joint Institute for Atmospheric Sciences (NJIAS)
E-mail: liuduanyang@cma.gov.cn;

**Manuscript #    NHESS-2023-230**

Manuscript Title: A New Method for Calculating Highway Blocking due to High Impact Weather Conditions

**Responses to Reviewer #3**

**General comments**

*In this study, the authors present a methodology to evaluate the effects of different weather-related events on the Chinese highway network. While the statistical and geographical description of weather events and its effect on road impact is very interesting and well presented, several concepts that seem fundamental to understand the results, such as highway load and losses, are not clear and can be misleading. Better care should be given to clearly explain these concepts and separate them from their usual denominations for traffic demands and direct economic impacts.*

*Overall, the value proposition (why this study and its results are important) is not clear. The study should be reevaluated to clearly provide a valuable discussion of its results, considering the strengths and limitations of the methodology. Large revisioning for the English language and grammar is required throughout the manuscript, which is beyond the scope of the present review.*
**RESPONSE: Thank you for your suggestions. The English language and grammar is required throughout the manuscript.**

**Specific comments**

**Abstract**

*The text in the abstract needs revision, I recommend to reorganize it considering a "why, how and what" storyline. "Why" focuses on the problem, "How" on the methodology and how it can help the problem, "What" focuses on the work done and the results. Currently, the Why is presented in a trivial way, the How is barely present and the What is too focused on the methods and not enough on the results and what they mean (the reference of 43% is not clear).*
**RESPONSE: Revised.**
**Abstract: Fog, rain, snow, and icing are considered to be the high-impact weather events often lead to the highway blockings, which in turn causes serious economic and human losses. At present, there is no clear calculation method for the severity of highway blocking which is related to highway load degree and economic losses. Therefore, there is an urgent need to propose a method for assessing the economic losses caused by high-impact weather events that lead to highway blockages, in order to facilitate the management and control of highways and the evaluation of economic losses. The goal of this work is to develop a method to be used to assess the high impact weather (HIW) effects on the highway blocking. Based on the K-means cluster analysis and the CRITIC (Criteria Importance through Intercriteria Correlation) weight assignment method, we analyze the highway blocking events occurred in Chinese provinces in 2020. Through cluster analysis, a new method of severity levels of highway blocking is developed to distinguish the severity into five levels. The severity levels of highway blocking due to high-impact weather are evaluated for all weather types. As a part of calculating the degree of highway blocking, a new method is proposed for China, and the highway load in each province is evaluated. The economic losses resulting from highway-blocking events caused by dense fog are specifically assessed, the highway losses caused by dense fog are mainly concentrated in Northern China, Eastern China and Southwestern China.**
* * *
**Introduction**

*The use of "high-impact weather conditions" is not ideal, it makes reference to impact and it is not clearly defined. I recommend focusing on the event and not the impact, adopting something similar to "adverse weather conditions". Also, it is best to not say "**the** high-impact weather conditions", just "high-impact weather conditions".*

**RESPONSE: Thank you for your suggestions. The World Meteorological Organization (WMO) defines high-impact weather as severe weather events that have significant adverse impacts on society, infrastructure, and the environment. These events can cause widespread damage, disruption, and loss of life (Marsigli, 2021).**

**High-impact weather refers to severe weather events that have a significant impact on human activities, infrastructure, and the environment. These events can cause widespread damage, disruption, and loss of life. In this study, we discuss the high impact weather that has an impact on highway operation.**
* * *
*Line 37*

*I recommend the following change: "Therefore, driving in foggy weather is a potentially dangerous activity **for users, which increases the potential for** road blocks (Yan et al. 2014)."*

**RESPONSE: Revised.**
* * *
*Line 54*

*the term HIW is used before it is described in line 57. Also, the study is not improving the effects on road blocking, but helping to better characterize and predict the impact. Please change the text to reflect that.*

**RESPONSE: Revised. In this respect, the current work will further help to improve HIW events prediction and the effects on road blocking.**
* * *
*Line 55*
*Statistics of China's road length and ranking needs a reference*
**RESPONSE: Revised.Liu, Dan, Liugang Sheng, and Miaojie Yu. 2023. "Highways and Firms' Exports: Evidence from China." Review of International Economics 31: 413-43.**
* * *
*Line 56*
*"The ability to estimate highway traffic demand caused by the highway blocking during adverse weather events; therefore, it is critically needed." needs to be rewritten to clarify. I recommend: "Therefore, there is a critical need to improve the ability to estimate highway traffic impact, caused by highway blocking during adverse weather events"*
**RESPONSE: Revised.Therefore, there is a critical need to improve the ability to estimate highway traffic, caused by the highway blocking during adverse weather events.**
* * *
*Line 57*
*Replace "factors" with "components" or "contributors"*
**RESPONSE: Revised.**
* * *
*Line 58*
*Replace "affected" with "caused"*
**RESPONSE: Revised.**
* * *
**Data and methods**

● *Please include a reference on where the authors are obtaining the data about the geometry and characteristics of the highway network.*
**Response: Revised. Highway blockage data comes from Highway Monitoring & Emergency Response Center, and the data recording process and specifications are in accordance with the "Information Reporting System for Highway Traffic Blockage of the Ministry of Transportation and Communications of the People's Republic of China" issued by the Ministry of Transportation and Communications of the People's Republic of China.**
* * *
● *In Section 2.3.3, it is unclear to me how the K-means algorithm is applied to the methodology proposed.*
**Response: Revised.**
**The correlations between data items are higher because some of the data items are obtained by performing calculations from other data items.Specifically, data normalization is firstly performed for all traffic**

$$Z = \frac{Z_i - Z_{min}}{Z_{max} - Z_{min}}$$

**indicators, where Z is the normalized data, calculated by** , **and then the weights are assigned to each normalized data by using the CRITIC weight method to obtain the weight value of each indicator. So we develop an equation (Eq. (6)) to calculate the degree of highway load.**
* * *
● *Please provide some lines into what is the physical meaning of the CRITIC weights in this context. This will help clarify the percentages reported in the results section.*

**RESPONSE: Equation3-4 and the key parameters are Revised.**

**Contrast intensity, expressed as a standard deviation, indicates the dispersion degree of an indicator. The larger the standard deviation is, the greater the dispersion degree is, the larger the differences between samples are, and the larger the assigned corresponding weights are. The standard deviation $S$ can be expressed in Eq. (3).**

$$S_j = \sqrt{\frac{\sum_{i=1}^{p}\left(x_{ij} - \frac{1}{n}\sum_{i=1}^{n} x_{ij}\right)^2}{n-1}} \tag{3},$$

**where $x_{ij}$ denotes the data processed by standard deviation, $S_j$ the standard deviation of the $j$th indicator, $n$ the total number of samples, and $p$ the total number of indicators.**
**Correlation is expressed as the correlation coefficient between indicators. The stronger the correlation between indicators is, the higher the repetition rate of information expression. Therefore, the corresponding weights of the indicators can be reduced to a certain extent. The correlation coefficient $R$ can be expressed in Eq. (4).**

$$R_j = \sum_{i=1}^{p}(1 - r_{ij}) \tag{4},$$

**where, $R_j$ indicates the correlation coefficients of the $j$th indicator with the other indicators, and $r_{ij}$ denotes the correlation coefficient of the $i$th indicator with the $j$th indicator.**
* * *
**Results**

● The bar plots in Figure 3.b need to be clarified to give a reference of what does the height of the bars represent.
**RESPONSE: Revised. Figure 3 High impact weather types leading to highway blocking in different provinces of China (a: proportion, b:   the height of the bars represent number:)**
* * *
● *Given the type of events considered (fog, rain, snow and ice) it is not clear to me why so many instances of road blocking are happening in the summer. Please include some lines to acknowledge and clarify this.*

RESPONSE: Revised. **Summer rainfall in mountainous areas is heavy, often accompanied by thunderstorms and short-term heavy precipitation events. Mountain terrain is special, the speed of rainwater runoff is fast, easy to form flash floods or debris flow, these natural disasters are the main reasons for blocking the highway.**
* * *
● *The way that the severity of the blocking is defined needs to be further clarified. It is not clear to me what "blocking mileage" (L) is. Maybe including a couple of illustrative examples would help.*

RESPONSE: Revised.

**Highway blocked miles are the total length of a highway that cannot be used normally during a given event or condition. For example, due to flash floods caused by precipitation in mountainous areas, there are 100 kilometers of highway can not be used normally, we believe that the blocked mileage caused by heavy rain is 100 kilometers.**
* * *
**Discussions**

● *The way that the concept of "highway load" is being treated in this section is not clear to me, or how it relates to losses. While the idea of using a "highway load" as a proxy for economic impact potential is clear, it is misleading to say they directly relate to economic losses. There are many other factors that come into play when evaluating economic losses. It is advised to clarify this concept, to assure the reader what the methodology is actually capturing and what the results mean potentially.*

RESPONSE: Revised. **We carry out the assessment of the carrying pressure per unit length of road in different areas by calculating the highway loading index, and when it carries more transportation tasks, more losses will be incurred in the event of a blockage.**
* * *
● *The results presented in Figure 8 should be first presented in the Results section, clearly explaining how they were derived and only then can they be discussed in this section. In the current format it is not clear where these values are coming from.*

RESPONSE: Revised.

**By carrying out this new calculation method proposed, we have analyzed and calculated all four types of catastrophic weather, and because the results are too lengthy, we have selected the fog weather blocking loss distribution, which has the highest percentage, for a demonstration. If needed, we can analyze the damage caused by the remaining three hazardous weather types of weather in a presentation and discussion.**

**In results part, the methods is**

● *Overall, this section lacks transparency about the strengths and limitations of the methodology.*

RESPONSE: Revised. **Overall, this method can effectively calculate and evaluate the economic losses caused by highway congestion caused by high impact weather events, so as to facilitate highway management and control and economic loss assessment.**

**However, The limitation of this approach is that the completeness of economic data in different regions can lead to bias in the evaluation results, and the evaluation model needs to be updated regularly to adapt to changing environmental and social needs.**

**Conclusions**

● *The final parts of this section makes reference to natural disasters, which have not been clearly discussed previously. I suggest this is removed to avoid misleading the readers into thinking that these events were accounted for in this study.*
**RESPONSE: Revised.**
* * *
● *There is a mention to direct and indirect losses, though their difference has not been discussed or defined in the paper, I suggest to remove it.*
**RESPONSE: Revised. The highway-blocking data used in this study is only for the year 2020 as a test year, and additional time series of observations are needed to validate the results. The assessment of losses is only judged by the degree of highway load. In the subsequent work, economic models can be employed to continue refining the research.**

---

## Referee Report (RR1)

Review of

**A New Method for Calculating Highway Blocking due to High Impact Weather Conditions**

Author(s): Duanyang Liu, Tian Jing, Mingyue Yan, Ismail Gultepe, Yunxuan Bao, Hongbin Wang, Fan Zu

Manuscript No.: nhess-2023-230

Manuscript type: Revised research article

Special issue: Indirect and intangible impacts of natural hazards

**General comments**

The **Revised manuscript** and **Responses to Reviewers** together addressed most of the reviewers' comments and the quality of the manuscript has been improved. However, some of the responses could have been included in the revised manuscript to further clarify the methodology and results. Moreover, the figure quality was reduced dramatically, compared to the original submission. The English writing was slightly improved.

The manuscript presents a study on a topic within the scope of the Natural Hazards and Earth System Sciences (NHESS). I would recommend this manuscript for publication after addressing the following minor comments.

**Specific comments**

1. **In general, the discussion and elaboration in the response to reviewers should be included in the main text**. Please further include some discussion in the responses to reviewers in the revised manuscript, for example:
   - Response to reviewer #1: Point 5 The CRITIC weigh method -> Bullet 2. The further consideration of weighting (e.g., conflicting nature of different indicators) should be included in the main text.
   - Response to reviewer # 2: Methodology - Point 3 "The use of highly correlated data is to better fit the CRITIC method. The CRITIC method combines two dimensions, intensity of comparison and conflict, to combine the weights of indicators. Comparative strength is expressed using the standard deviation, which reflects the degree of dispersion of data within the indicator, and conflict is expressed using the correlation coefficient, which reflects the correlation between indicators."
   - Response to reviewer #3: Results – point 2 "Summer rainfall in mountainous areas is heavy, often accompanied by thunderstorms and short-term heavy precipitation events. Mountain terrain is special, the speed of rainwater runoff is fast, easy to form flash floods or debris flow, these natural disasters are the main reasons for blocking the highway."
   - And more.

2. **Figure resolution was reduced significantly.**
   - Please increase the resolution of all figures
   - Please include the longitude, latitude and legend of all the geographic Figure 1 and 3
   - Figure 1a and 1b can indicate the regional division using different colour codes and then add a region colour legend while keeping the province labels in 1b
   - Figure 3b the scale of the vertical bar was missing in the legend

3. WMO definition of "High-impact weather" was included in the revised manuscript. However, the term "adverse weather event" was also used. It is recommended to use consistent terminology.

**Minor comments**

4. "response time in different high-impact weathers **asl** the input vectors in the calculations." Should "asl" be "as"?
5. Line 49-50: "The weather forecasting products, if used, the road transportation sectors can generate great economic benefit." Should be "**If** the weather forecasting products **are used**, the road transportation sectors can generate great economic benefit"
6. Line 91-93: "it can be divided into two categories: planned and **unexpected**." and "**Sudden-type** blockages include…", please use the consistent category label "unexpected" or "sudden-type"

---

## Author Response (AR2)

Dear Editor and Reviewers,

Thank you very much for the second-round constructive suggestions on our paper entitled "A New Method for Calculating Highway Blocking due to High Impact Weather Conditions". They are very helpful for improving the manuscript. We have revised the manuscript accordingly. Thanks for consideration.

Yours sincerely,

Duanyang Liu (Corresponding Author)

Nanjing Joint Institute for Atmospheric Sciences (NJIAS) E-mail: liuduanyang@cma.gov.cn;

**Manuscript # NHESS-2023-230**

Manuscript Title: A New Method for Calculating Highway Blocking due to High Impact Weather Conditions

Responses to Reviewer #1 Round 2

**General comments**

*The Revised manuscript and Responses to Reviewers together addressed most of the reviewers' comments and the quality of the manuscript has been improved. However, some of the responses could have been included in the revised manuscript to further clarify the methodology and results. Moreover, the figure quality was reduced dramatically, compared to the original submission. The English writing was slightly improved.*

*The manuscript presents a study on a topic within the scope of the Natural Hazards and Earth System Sciences (NHESS). I would recommend this manuscript for publication after addressing the following minor comments.*

**RESPONSE: Your points are considered carefully and text is revised. Thank you for your suggestions. All figures have been improved and figures' quality are checked. It was a copy previously.**

\-----------------------------------------------

**Specific comments**

*1*   **In general, the discussion and elaboration in the response to reviewers should be included in the main text**. *Please further include some discussion in the responses to reviewers in the revised manuscript, for example:*

- *Response to reviewer #1: Point 5 The CRITIC weigh method -> Bullet 2. The further consideration of weighting (e.g., conflicting nature of different indicators) should be included in the main text.*

- *Response to reviewer # 2: Methodology - Point 3 "The use of highly correlated data is to better fit the CRITIC method. The CRITIC method combines two dimensions, intensity of comparison and conflict, to combine the weights of indicators. Comparative strength is expressed using the standard deviation, which reflects the degree of dispersion of data within the indicator, and conflict is expressed using the correlation coefficient, which reflects the correlation between indicators."*

- *Response to reviewer #3: Results – point 2 "Summer rainfall in mountainous areas is heavy, often accompanied by thunderstorms and short-term heavy precipitation events. Mountain terrain is special, the speed of rainwater runoff is fast, easy to form flash floods or debris flow, these natural disasters are the main reasons for blocking the highway."*

- *And more.*

**RESPONSE: Thank you for your suggestions. The discussion section is improved to response for your points and shown in the text now.**

\-----------------------------------------------

**2    Figure resolution was reduced significantly.**

● Please increase the resolution of all figures.

● Please include the longitude, latitude and legend of all the geographic Figure 1 and 3.

● Figure 1a and 1b can indicate the regional division using different colour codes and then add a region colour legend while keeping the province labels in 1b.

● Figure 3b the scale of the vertical bar was missing in the legend.

**RESPONSE: The figures have been revisited for your points and the current figs quality is improved significantly.**
* * *
*3    WMO definition of "High-impact weather" was included in the revised manuscript. However, the term "adverse weather event" was also used. It is recommended to use consistent terminology.*

**RESPONSE: We agreed with your point and now we used only "*High-impact weather*"**
* * *
**Minor comments**

1. *"response time in different high-impact weathers **asl** the input vectors in the calculations." Should "asl" be "as"?*

**RESPONSE: revised, thank you.**
* * *
2. *Line 49-50: "The weather forecasting products, if used, the road transportation sectors can generate great economic benefit." Should be "**If** the weather forecasting products **are used**, the road transportation sectors can generate great economic benefit"*

**RESPONSE: revised as you suggested, thank you.**
* * *
3. *Line 91-93: "it can be divided into two categories: planned and **unexpected**." and "**Sudden-type** blockages include…", please use the consistent category label "unexpected" or "sudden-type".*

**RESPONSE: revised, now it is used as "unexpected", thank you.**
* * *
Responses to Reviewer #2 Round 2

**General Comment**

   *Dear Authors, I went through the revised version of the manuscript, finding it slightly improved since the previous version. Replies and modifications partially addressed previous comments. In the cases where no modifications have been done there are no significant and substantial justifications.*

   *Thus, in line with the previous review, I consider the manuscript potentially suitable for publication, but only after having properly addressed some issues.*

   *I recommend the Authors to carefully deal with such comments, modifying accordingly the manuscript or justifying their choices.*

**RESPONSE: We went over your previous and current suggestions, and improved the text coherently. Thank you for your suggestions. The figures have been redone and their quality is improved significantly.**
* * *
**Specific comments:**

*1.  - L108-L115: there are some repetitions here, please revise.*

**RESPONSE: revised, thank you. The dataset contains 16 indicators: province name, submitting department, route name, route number, starting and ending pile number, reasons of highway blocking, blocking mileage, status, blocking type, information event classification, site description, disposal measure, time of finding blockage, submitting time, expected recovery time, and actual recovery time. (There are two nouns that need to be explained in detail: firstly, Highway pile numbers are usually combined with the milestone system and are expressed in K kilometers ± meters. That is, along the direction of the road, the pile number at the starting point is k0+000, and one pile number is marked every certain distance (such as 100 meters), and the corresponding place is marked.   Second, the blocking mileage is the distance of the highways blocking, for example, due to flash floods caused by precipitation in mountainous areas, there are 100 kilometers of highway can not be used normally, we assume that the blocked mileage caused by heavy rain is 100 kilometers.) Since highway blocking information is submitted by manual statistics, there is a possibility of manual statistical errors. Therefore, all data were pre-corrected with a time series correction and then verified based on the cause of the blockage and the meteorological data of the station at the time, among other things. Quality control resulted in the retention of 95% of valid data.**
* * *
*2.  - L120: we "assume" instead of "we believe".*

**RESPONSE: Considering your point, text is revised, and now much improved.**
* * *
*3.  -L119-L121: please revise the text and check for typos.*

**RESPONSE: Text is revised, and grammatically improved.**
* * *
*4.  Figure 3: Sorry, but I recall here the same comment previously reported with regards figure 3, since no replies and actions were reported/taken in the current version and Authors' reply. Figure 3 is too small. I would consider to separate the two panels. Here the comment: "Region names should be added to the map. Plots, as well as legends, should be more quantitative. Which are the values associated to the bars in pale 3b? Also, I suggest removing the lateral minor-boxes showing the islands, since it looks like there are nor results for those regions."*

**RESPONSE: Fig 3 is redone based on your point, and all other figures have been revisited.**
* * *
*5.  - In general, Figures are of poor quality, making difficult to read labels and results.*

**RESPONSE: All figures have been redone or modified based on the Revs' comments.**
* * *
*6.  In my opinion, all my previous comments on providing a deeper discussion on the methodology and results were not properly addressed. I recall the aspects risen in the previous review, which I believe was not properly addressed by the Authors in this new version.*

**RESPONSE: Based on your points, now both discussion and method sections are revisited to include your comments, and we believe now that the text is improved significantly.**
* * *
7. - *L345-347 (L386-387): "I disagree with the use of losses or damages, as adopted in the discussion and in Figure 8. From what I have understood, this analysis is an exposure evaluation rather than a damage assessment. The classifications shown in Figure 7 (which can be further explained with the details of Figure 8) provide a picture of the overall highway loads over the area, which do not necessarily correspond to real damage. This is a key aspect that deserves consideration and affects the overall scope and ambition of the manuscript. Furthermore, Conclusions should be revised accordingly to such definition."*

**Response : We agreed with your point, and this is clarified in the text now (4.1 Economic Losses exposure evaluation due to fog-related highway blocking; Figure 8. Economic losses exposure evaluation caused by highway blocking due to fog). Based on this point as suggested we modified the conclusions.**
* * *
*Also, Reviewer#3 similarly commented on this topic.*
8. - *The overall discussion is lacking, with insufficient critical analysis of the strengths and limitations of the methodology.*

**Response : Discussion section considering of the strength and limitations of the methodology is modified and provided below.**

**4. Discussions about the strengths and limitations of this methodology**

[revised manuscript text omitted]

Overall, our analysis based on Eq. 8 can effectively estimate and evaluate the economic losses caused by highway congestion owing to high-impact weather events. However, the limitation of our approach is that the lack of economic data in different regions can lead to bias in the results. Therefore, the evaluation model in the present study needs to be updated regularly to adapt to changing environmental and social conditions.
* * *
9. - The weights shown in eq. 8 are quite uniform among the considered items. What is the sensitivity of the outcomes to variations in such weights? Any comments on their values? It would have been beneficial to consider additional data or to remove some of those already considered.

**Response : We evaluated your points and explained below. Note that sensitivity of outcomes to variable weights are critical and important, this is emphasized now. It is found out that 20% change of one the weights resulted in the outcomes at about 30%, suggesting that your point is very important and this is discussed now. Thanks for this critical point.**

**Using Eq. 8 and the results of Table 6 and Table 7, it can be seen that the weights shown in eq. 8 are quite uniform among the considered items. This is primarily due to our data normalization of the relevant economic indicators. Through the analysis of Economic Losses due to Fog-Related Highway Blocking conditions in Section 4.1. We can also see that the differences between Figures 7 and 8 comes from mainly variations in economic indicators foreach province, leading to different economic loss values, it is found out that 20% change of one the weights resulted in the outcomes at about 30%. And thus, the sensitivity of the outcomes to variations become important. Additionally, different provinces experience various high-impact weather types in different seasons, resulting in corresponding variations in economic indicators and economic losses.**

**In reviewing previous research, few have mentioned the assessment of disasters caused by high-impact weather on highway traffic losses. Because of this issue, our work primarily using data mining analysis, explores the relationship between high-impact weather impact on economic losses**

related to highways. The weight coefficients are derived entirely from high-impact weather and economic data without any human intervention. Due to the large geographical area covered by the data used in this study and the short time series, the observed differences are relatively small, but the results vary across different provinces. It is hoped that future research will further explore the sensitivity of these indicators over much smaller regions. Therefore, after weighing the pros and cons, we chose to set the weight coefficients relatively close to each other to reflect the relative importance of these indicators in the overall assessment. On the other hand, if the analysis were performed over much smaller regions, the results would be the strong function of those weight values and this will be considered in a future work.